# An Update on Current Antiviral Strategies to Combat Human Cytomegalovirus Infection

**DOI:** 10.3390/v15061358

**Published:** 2023-06-12

**Authors:** Kingshuk Panda, Deepti Parashar, Rajlakshmi Viswanathan

**Affiliations:** 1Dengue-Chikungunya Group, Indian Council of Medical Research-National Institute of Virology, Pune 411001, India; 2Bacteriology Group, Indian Council of Medical Research-National Institute of Virology, Pune 411001, India

**Keywords:** antiviral, HCMV, RNAi, CRISPR-Cas, TALEN, aptamer

## Abstract

Human cytomegalovirus (HCMV) remains an essential global concern due to its distinct life cycle, mutations and latency. As HCMV is a herpesvirus, it establishes a lifelong persistence in the host through a chronic state of infection. Immunocompromised individuals are at risk of significant morbidity and mortality from the virus. Until now, no effective vaccine has been developed to combat HCMV infection. Only a few antivirals targeting the different stages of the virus lifecycle and viral enzymes are licensed to manage the infection. Therefore, there is an urgent need to find alternate strategies to combat the infection and manage drug resistance. This review will provide an insight into the clinical and preclinical antiviral approaches, including HCMV antiviral drugs and nucleic acid-based therapeutics.

## 1. Introduction

Human cytomegalovirus (HCMV) is an ubiquitous virus either existing quiescently in healthy persons or causing severe disease in immunocompromised patients and congenitally infected newborns. To date, eight herpesviruses have been identified that are known to have humans as primary hosts. According to ICTV (International Committee on Taxonomy of Viruses), they have been named as Human herpesvirus 1–8 (HHV 1–5, 6A, 6B, 7, 8) [1]. HCMV is officially known as Human herpesvirus 5. HCMV is characterized and classified as a prototypic beta-herpesvirus [2] and shares common properties with other beta-herpesviruses, including appearance in electron micrographs, prolonged reproductive cycle, species specificity and tropism for differentiated hematopoietic and epithelial cells [3]. After primary infection in immunocompetent hosts, HCMV rarely causes disease. Infection may be asymptomatic or result in mild mononucleosis and flu-like symptoms. However, the virus is not completely cleared from the system and can establish a persistent or latent infection in the host. Reactivation of HCMV occurs when the virus overcomes the innate immune response resulting in asymptomatic infections, mild, moderate or severe disease. This depends on several variables, the most important being the immune status of the patient [4]. HCMV disease can mimic several other conditions and lead to significant diagnostic challenges [5]. It is considered a critical health concern for high-risk populations like HIV/AIDS patients, transplant recipients and congenitally infected infants who bear the brunt of infection with HCMV [6]. HCMV occurrence in a population shows an age dependent rise in seroprevalence. Seroprevalence of more than 60% is reported in adults from developed countries and more than 90% in low and middle income countries [7]. However, some developed countries, such as Japan, Sweden and Taiwan, have reported high HCMV seroprevalences [8]. Infection is more common in those from lower socio-economic groups and from non-Caucasian backgrounds [8]. About 60–90% of all renal transplant candidates have latent HCMV infection; however, 20–60% develop symptoms and active infection [9]. HCMV is the most common cause of congenital infection, affecting 0.1–2% of babies born across the world and can result in substantial neonatal and childhood morbidity due to hearing loss, visual impairment and developmental delay [10].

HCMV is a large DNA virus that belongs to the genus of Cytomegalovirus and the betaherpesvirinae subfamily [11]. The virus contains a double-stranded DNA genome of approximately 235 kbp, and it encodes numerous functional proteins, microRNAs, long-non coding RNAs and small peptides [3]. The HCMV genome contains >200 predicted open reading frames, with 71–77 proteins identified by mass spectrometry and in silico analysis [12,13,14]. Additionally, it also ex-presses four major non-coding RNAs (lncRNAs) along with at least 16 pre-miRNAs and 26 mature miRNAs [15]. The genome contains several completely unknown genes yet to be characterised.

The large DNA virus is arranged in a class E structure, with two unique regions (unique long UL and unique short US). These two regions are both flanked by a pair of inverted repeats (terminal repeat long TRL and terminal repeat short TRS). The TRL and TRS share a region of a few hundred bps which is called “a sequence”; the other regions of the repeats are usually referred to as “b sequence” and “c sequence”. The genome exists as an equimolar mixture of four genomic isomers by inversion of UL and US regions [16]. There are 162 capsomeres in the capsid that surrounds the genome, forming a helix-like DNA core [17,18]. The 100 nm- diameter capsid in turn, is sur-rounded by the tegument. The virion is enveloped by lipid bilayers containing viral glycoproteins, resulting in an infectious virus particle with a diameter of around 180 nanometres [19].There are several types of glycoprotein complexes (gC) contained in the virus that are essential for infection of the host; gC-I contains homotrimers of gly-coprotein B (gB), while gC-II is composed of glycoprotein M (gM) and N (gN), which contribute to the initial binding to the cell membrane, and immune evasion [20,21,22]. The gC III glycoprotein complex consists of either the trimer gH/gL/gO or the pentameric complex, gH/gL/pUL128/pUL130/pUL131. The pentameric complex is essential for HCMV entry into endothelial and epithelial cells and is a target for neutralizing antibody responses [23,24]. The virus shows a wide range of cellular tropism and can cause opportunistic infections in a variety of organs and tissues in the body [25]. The gastrointestinal tract, liver, and central nervous system are the primary site for symptomatic infection. HCMV can remain latent in secretory glands, lymphoreticular cells, kidneys, and other tissues as well [25]. It is a classic opportunistic pathogen and can cause severe infections in immunocompromised individuals, such as persons suffering from advanced human immunodeficiency virus (HIV) infection/Acquired Immunodeficiency Syndrome (AIDS) and transplant recipients [26]. Once termed the “troll of transplantation” because it threatened solid organ transplant recipients and negatively impacted morbidity, mortality, and transplant outcomes. HCMV is now considered the “troll of tolerance” in the context of transplantation, due to its ability to interfere with the establishment and maintenance of immunological tolerance, which is necessary for successful transplantation outcomes [27]. Latent HCMV may also be reactivated in critically ill patients (especially elderly) in intensive care units [27,28]. Very low birth weight (VLBW) and premature babies are at risk, albeit low, of acquiring HCMV from breast milk, leading to development of postnatal HCMV infection, which may present as sepsis like syndrome [29].

Furthermore, the emergence of drug resistance and dose-limiting toxicities limit anti-viral treatments for HCMV infections and diseases. In addition, there are no approved vaccines or immunoglobulins available to prevent congenital HCMV infection. However, intravenous immunoglobulin (IGIV) preparation such as Cytogam have been used to treat certain HCMV infection, particularly in immunocompromised patients such as transplant recipients [30]. The use of CMV IG as a rescue therapy in cardiothoracic transplant recipients has been shown to be safe, well tolerated and effective [31]. RG7667, a combination of two monoclonal antibodies was evaluated in kidney trans-plant recipients. It was well tolerated, numerically reduced the incidence of CMV infection within 12 and 24 weeks posttransplant, delayed time to CMV viremia, and was associated with less CMV disease than the placebo [32]. There are several unmet medical needs for HCMV infections, which calls for the development of new, safe, and effective anti-HCMV compounds with novel mechanisms of action. This review discusses the updated information on possible antiviral strategies available to combat HCMV infection.

## 2. Antiviral Drugs against HCMV

Currently, only a few antiviral drugs are approved for HCMV infection (Table 1).

### 2.1. Ganciclovir and Valganciclovir

Ganciclovir (GCV) or its prodrug valganciclovir (VGCV), an acyclic nucleoside analog, is the preferred antiviral agent for the treatment of confirmed HCMV disease. In 1987, Erice et al. first reported that ganciclovir showed antiviral activity against HCMV in patients with depressed immunity and bone marrow and organ transplant recipients [33]. Phosphorylation to ganciclovir monophosphate is mediated by pUL97, a viral kinase, followed by phosphorylation to ganciclovir triphosphate by cellular kinases [34]. Once activated, ganciclovir -TP is incorporated into the viral DNA during replication and acts as a chain terminator by blocking the addition of nucleotides to the growing DNA chain. GCV is available in intravenous and oral formulations and is also used intravitreously for treating HCMV retinitis in HIV/AIDS patients [33]. In congenital CMV infections, although structural CNS defects cannot be reversed, GCV therapy is important for limiting the developmental delay [35]. Studies have shown that GCV can reduce or stabilize the hearing impairment and improve neuro-developmental outcomes in infants with congenital CMV when treated within the first 4 weeks of life [36,37,38]. VGCV, a valine ester of GCV, is an oral pro drug and plays a major role in HCMV prevention in immunocompromised hosts. Treatment is indicated and of proven efficacy in severe, life-threatening or disseminated disease, including evidence of central nervous system (CNS) involvement, hepatitis, pneumonia and thrombocytopenia [39,40]. VGCV treatment for six months as compared to six weeks was shown to modestly improve long term hearing and developmental outcomes in infants with symptomatic congenital CMV [37]. It has the advantage of being an oral drug, which improves ease of administration [41]. However, drug toxicity remains a major problem for both GCV and VGCV. It is important to estimate baseline renal function before initiating therapy and to monitor for evidence of bone marrow suppression and nephrotoxicity during therapy with either of these drugs.

### 2.2. Cidofovir

Among nucleotide analogs, cidofovir (CDV), which is administered intravenously, was the first to receive FDA approval. It is mostly used to treat HCMV infection in patients with AIDS and in transplant recipients. CDV is a monophosphate nucleotide analog [42]. Through cellular phosphorylation, it is converted into diphosphate (active) form and competitively inhibits the incorporation of deoxycytidine triphosphate (dCTP) into viral DNA by viral DNA polymerase [43]. As a result, it disrupts further chain elongation. Cidofovir is unaffected by a mutation in the HCMV-encoded phosphotransferase (pUL97) that confers resistance to GCV. It has been reported that patients with AIDS who received immediate CDV therapy exhibited less progression of HCMV retinitis than those who deferred treatment [44]. If a patient experiences a relapse of HCMV retinitis after previous treatment, CDV is generally used again as a second-line treatment option; CDV, as part of combination therapy, was reported to show comparable efficacy to GCV or foscarnet in haplo-HSCT recipients (patients with haploidentical hematopoietic stem cell transplantation) with HCMV infection [45]. However, CDV has the disadvantage of being nephrotoxic. This is mainly because of its comparatively slow secretion into the renal tubular lumen, in contrast to uptake from the blood. This persistence of the drug in the proximal renal tubular cells results in nephrotoxicity [46].

### 2.3. Foscarnet

Foscarnet is an inorganic pyrophosphate with a broad spectrum of activity. The drug works by binding reversibly near the pyrophosphate-binding site of viral DNA poly-merase pUL54 and prevents the cleavage of pyrophosphate from the deoxynucleotide triphosphate, in turn halting chain elongation. Two important uses of foscarnet are in the early stage following hematopoietic stem cell transplant due to comparatively less suppression of bone marrow than GCV; and in treating GCV-resistant HCMV infection in transplant recipients and HIV/AIDS patients. It is available as an IV formulation and like GCV, it is also used intravitreously for treatment of HCMV retinitis.

### 2.4. Letermovir

Letermovir is FDA approved for prophylaxis of HCMV infection and disease in adult HCMV-seropositive recipients [47]. It is prepared as intravenous as well as oral formulations. The terminase complex is a molecular machine that plays a critical role in the HCMV replication cycle. It is important for cleavage and packaging of viral DNA into capsids, which occurs in the nucleus after the viral DNA is replicated in the nucleus. The terminase complex is composed of two subunits, pUL56 and pUL89, which work together to cleave the viral DNA. This drug targets the HCMV terminase complex, specifically the ATPase pUL56 subunit and prevents it from cleaving the viral DNA. This inhibition of the terminase complex blocks the packaging of the viral DNA into capsids, which in turn prevents the formation of infectious virus particle [48]. The use of leteromovir for HCMV prophylaxis or treatment in heart and lung transplant recipients was first examined in 2014 [49]. The study reported that letermovir was well tolerated with only minor side effects. Although, the development of HCMV DNAemia on prophylaxis was observed in a few patients, possibly due to uncertain doses. Later, it has also been clinically applied for HCMV prophylaxis and treatment in HSCT recipients [50].

### 2.5. Maribavir

Maribavir is a potent, selective, orally bioavailable benzimidazole riboside that is active against CMV infection in humans. This drug works by inhibiting the UL97 kinase enzyme, which plays a critical role in replication by phosphorylating proteins in-volved in the assembly of the viral capsid [51]. Maribavir inhibits this enzyme and prevents the phosphorylation of these proteins, which in turn disrupts the assembly of the viral capsid. This leads to the inhibition of HCMV replication and the production of infectious viral particles. Phase III clinical trial data suggests that Maribavir shows better viremia clearance without significant side effects [52]. Recently, Maribavir was approved in the USA for the treatment of post-transplant CMV infection/disease that is refractory to treatment with ganciclovir, valganciclovir, cidofovir or foscarnet [53].

### 2.6. Repurposed Drugs

Recently, a few neuroactive mood stabilizers are reported to be potent antivirals against HCMV. Valproate is a widely prescribed anti-epileptic drug employed to treat multiple psychiatric and neurological diseases. A study by Ornaghi et al., first reported that Valpromide (VPD) and Valnoctamide (VCD) can inhibit both human and murine CMV both in vitro or in vivo [54]. The study confirmed the drug’s efficacy in blocking CMV in vivo and reduction in viral load in target organs without any side effects. Both drugs target CMV at an earlier stage of infection, specifically during viral attachment to cell surface heparan sulfate proteoglycans (HSPGs). The exact mechanism of action is still unreported, although the inhibition of HCMV attachment may be due to a reversible interaction with either HSPGs or free virions [54].

Leflunomide is a pyrimidine synthesis inhibitor belonging to the DMARD (disease-modifying antirheumatic drug) class of drugs originally indicated for treating rheumatoid arthritis. It is cheap and easily available and shown to have anti-HCMV properties both in vitro and in vivo. It also has been used effectively for HCMV which has reactivated after renal transplant [55] as well as refractory HCMV infection in allogenic stem cell transplant recipients, particularly when the copy number is low [56].

A critical issue with all of these treatments is that HCMV can develop resistance to antiviral drugs, especially in immunocompromised individuals, including AIDS patients and those who have received prolonged antiviral treatment. In several cases of HCMV treated with GCV/VGCM, antiviral drug resistance has also been reported. Mutation in the viral protein kinase gene (UL97) and the DNA polymerase gene (UL54) confers resistance to GCV/VGCV (Figure 1) [57,58,59]. Therefore, there is an urgent need to develop a new antiviral with higher efficacy and fewer side effects.

## 3. Cytomegalovirus Vaccines: Current Status and Future Prospects

HCMV is a complex virus that has evolved many mechanisms to evade the immune system, making it challenging to develop an effective vaccine. In order to prevent infection and its associated costs, it has been considered the highest priority to develop a vaccine against CMV. HCMV vaccines were first developed in the 1970s when AD169 and Towne strains were attenuated to function as active immunity boosters [60,61]. The initial results were promising, but further analysis revealed that protection against the infection was not significant. The most severe consequences of contact with HCMV are experienced by children born with congenital infections and immunocompromised individuals. Thus, although target populations remain controversial, pregnant women and women of childbearing age are more likely to be suitable as also patients receiving transplants. In immunocompromised individuals, the major challenge is generating adequate immunity through vaccination. This target population may benefit from an HCMV vaccine that stimulates both T cells and neutralizing antibodies. Since many women are infected by their children or by jobs involving close contact with children, vaccines for toddlers may provide indirect protection [62].

Despite these challenges, various vaccine candidates are currently under clinical trial, including live-attenuated vaccines, subunit vaccines, viral vector vaccines, chimeric peptide vaccines, vaccines based on enveloped virus-like particles, plasmid-based DNA vaccines, RNA-based vaccines and peptide vaccines [63]. Current vaccine candidates have focused on a few antigens; neutralizing antibody targets, such as gB, gH and PC; and T cell epitopes, such as pp65 and IE1. Initially, gB seemed a perfect choice, but trials showed limited efficacy [64]. A critical step for developing an HCMV vaccine is the identification of novel potent epitopes/antigens and cellular receptors implicated in virus entry into host cells. This new awareness has led to investigation of the humoral response to HCMV infection to understand which viral antigens could be more important to induce protective antibodies and be used in a vaccine.

## 4. Possible Alternative Strategies to Combat HCMV Infection

### 4.1. RNAi-Based Therapeutics against CMV

RNA interference is a mechanism by which cells can silence specific genes. Small interfering RNAs (siRNAs) and microRNAs (miRNAs) are the two main categories of small RNAs investigated extensively for their antimicrobial activities and multiple roles in regulating gene expression [65]. Both the siRNA and miRNA exert gene silencing in a post-transcriptional manner [66]. The first step of the RNA interference mechanism is the entry of dsRNA into the cytoplasm of a cell siRNA specifically targeting and degrading mRNA molecules that are complementary to the sequence of siRNA. miRNA, on the other hand, binds to specific sides on the 3**′**-untranslated region (3**′**-UTR) of the mRNA molecules, inhibiting their translation into protein. An 18–25 nt synthetic siRNA/miRNA strand could specifically bind to a target region and cause gene silencing [67] (Figure 2). A few studies have reported the inhibition of HCMV by both siRNA and miRNA. An early in vitro study by Xiaofei et al. [68] had designed a siRNA, named siX3 to target the coding sequences within shared exon 3 of UL123 and UL122 transcripts encoding IE1 and IE2 immediate-early proteins of HCMV. The study reported reduced levels of viral protein expression, DNA replication and progeny virus production upon pre-treatment with the siRNAs.

Comparatively few studies are available to understand miRNA-based antiviral strategies against HCMV. In one such in vitro study the role of miRNA in regulating the replication of HCMV has been investigated. The study focuses on the viral protein IE2 and its interaction with host miRNAs, such as miR-142 and miR-93 [69]. It was suggested that miRNA can regulate the expression of HCMV genes and thus control the replication and spread of the virus and therefore could be considered an alternative antiviral strategy to control viral replication [69].

### 4.2. Ribozyme-Based Therapeutics

The ribozyme-based antiviral strategy involves ribozymes, which are RNA molecules with catalytic activity. This approach to antiviral therapy offers several potential advantages over other strategies, including specificity, reduced toxicity and the ability to target multiple sites on the viral RNA [70]. Ribozymes-based antiviral strategies involve the delivery of ribozymes to infected cells through exogenous delivery or endogenous delivery. Exogenous delivery involves directly introducing ribozyme into cells through a delivery vehicle, whereas endogenous delivery involves expressing ribozyme within cells using plasmids or viral vectors [71]. Once inside the cells, the ribozymes can bind to and cleave the viral RNA, preventing replication of the virus [72]. The ribozyme ribonuclease P (RNase P) is a unique RNase as it is an RNA catalyst similar to a protein enzyme [73].

It functions by cleaving the precursor sequence of RNA on the tRNA molecule and stopping the protein synthesis (Figure 2). Nowadays, ribozyme-based strategies have been considered a vital gene-silencing tool for viral infection [74]. An early study constructed a sequence-specific ribozyme (M1GS RNA) using the catalytic RNA subunit from RNAse P of *Escherichia coli* to target the shared exon 3 of the major immediate early mRNAs (UL122-123) of HCMV [75]. In vitro analysis reported more than 80% reduction in the expression level of IE1 and IE2 protein and 150-fold virus reduction upon comparison with the control. IE1 and IE2 share 85 amino-terminal amino acids because of alternative splicing and polyadenylation of transcripts initiating at a strong promoter enhancer. Therefore, targeting the overlapping region of the mRNAs coding for IE1 and IE2 simultaneously shuts down the expression of both proteins along with effective inhibition of viral replication. In 2004, Kim et al. [76] had constructed a functional ribozyme named M1GS RNA that targeted the overlapping mRNA region of two HCMV capsid proteins, the capsid scaffolding protein (CSP) and assemblin. HCMV-infected cells expressing the functional ribozyme showed an 85% reduction in CSP and assemblin expression and an 800-fold reduction in viral growth. Yang et al., 2014 reported that a variant of RNase P ribozyme can target the HCMV essential transcription IE2. In vitro, the variant RNase P ribozyme was efficient to cleave the HCMV IE2 mRNA [77]. Deng et al. [78] reported that ribonuclease P-associated external guide sequences (EGS) variants effectively hydrolysed target mRNAs encoding HCMV major capsid proteins. Comparing engineered EGS with natural tRNA-derived EGS, the study confirmed that engineered EGS cleaves mRNA more efficiently. Upon treatment with engineered EGS, the viral growth was inhibited by 10,000-fold, thus demonstrating higher efficiency in blocking the expression of HCMV genes and viral growth. In all the reports, the result suggests that the ribozyme does not interfere with host gene expression and does not exhibit any cell cytotoxicity under both in vitro and in vivo conditions. However, the successful delivery of the designed ribozyme is a clinical challenge.

### 4.3. CRISPR/Cas9-Based Therapeutics

Clustered Regularly Interspaced Short Palindromic Repeats (CRISPR)-based antiviral therapy is a promising technology that has the potential to revolutionize the treatment of viral infection [79]. Several CRISPR systems can be used as therapeutic agents by directly cleaving DNA and RNA viral genomes in a targeted and easily adaptable manner (Figure 2). The CRISPR/Cas9 system consists of a guide RNA complementary to a specific target sequence in the viral genome and the Cas9 endonucleases, which cleave the viral DNA at the target site [80]. By cleaving the viral DNA, the CRISPR/Cas9 system can inactivate the viral genes and prevent virus replication.

Two CRISPR/Cas9 strategies have been used to target the UL122/123 gene in the HCMV genome, which is essential for the regulation of the lytic cycle and reactivation from latency [81]. Two different strategies were used in the study, a single plex strategy which consists of one gRNA to target the start codon and a multiplex strategy, which consists of three gRNAs to target the complete UL122/123 gene. The multiplex strategy excised 90% of the immediate early (IE) gene in all viral genomes and inhibited IE protein expression.

Previously, Van Diemen et al. also tried to combat the latent and active HCMV infection by introducing CRISPR/Cas9 technology targeting essential viral genes [82]. The study reported that abrogation of HCMV replication was observed in anti-HCMV sgRNA cells targeting the essential genes UL57 and UL70 upon comparison with the sgRNAs targeting the non-essential genes US7 and US11. This in vitro study provides early clues that CRISPR/Cas9 system could be considered for developing antiviral strategies to impair HCMV replication.

In another study the CRISPR/Cas9 system was used to target the IE region of the HCMV genome via sgRNAs [83]. The study confirmed that, after infection with CRISPR/Cas9/sgRNA lentiviral constructs, viral gene expression and virion production are reduced in HFF primary fibroblasts [83]. Interestingly, in the THP-1 monocytic cell line, inhibition of viral DNA production and reactivation was also observed.

CRISPR/Cas9-based antiviral strategies for HCMV are still in the early stages of development, but they hold promise as a new approach to antiviral therapy. Further research is needed to fully understand the efficacy and safety of CRISPR/Cas9-based antivirals and to identify the best methods for delivering the CRISPR/Cas9 system to infected cells.

### 4.4. TALEN-Based Therapeutics

Genome modification platforms are becoming a valuable tool for developing novel therapies against severe viral infection. Transcription Activator-like Effector Nucleases (TALEN) are generally synthesized by designing DNA binding domains that recognize a specific DNA sequence and fuse this domain to a nonspecific DNA cleaving domain [84]. TALENs have become recognized as an effective genome editing tool for their low toxicity and high efficacy. TALEN-based antiviral strategies have shown promise in laboratory studies with MCMV but are still in the early stages of development [85]. Further research is needed to completely understand the efficacy and safety of TALEN-based antivirals in HCMV.

### 4.5. Aptamer-Based Therapeutics

Aptamers are single-stranded DNA, RNA or peptide molecules that can be engineered to bind to the specific target molecule [86]. Aptamers can be designed to bind to and inhibit critical components of the virus, such as viral protein or RNA [72]. Aptamers have previously been used to target proteins required for virus entry into the host cells, such as cellular receptors and viral envelope proteins [87,88].

In 2009, the aptamer-based gene silencing technology was tested against HCMV for the first time (Figure 2) [89]. The UL84 open reading frame of HCMV encodes an essential protein for lytic viral replication. The study used peptide aptamer technology to interfere with the nuclear import pathway of pUL84. It was confirmed that several peptide aptamers from a randomized peptide expression library bind to the unconventional pUL84 NLS under intracellular conditions. Upon treatment, around 50–60% of decreased viral replication was observed in primary human fibroblast cells expressing pUL84-specific aptamers. A 50–70% reduction of viral plaque formation was also observed, along with 70–90% inhibition of virus release in the presence of pUL84-specific aptamer [89]. Aptamers have several advantages as antiviral agents, such as specificity and stability, and can be easily produced in large quantities. In addition, they can be delivered into infected cells through various routes, such as intravenous injection or topical application. The development of aptamer as an antiviral agent for HCMV is still in the very early stages. More research is needed to solve the difficulties, such as target-specific delivery and stability.

Antisense molecules are promising gene-targeting technologies for the treatment of various viral infections. Fomivirsen, an antisense oligonucleotide-based drug, was developed to treat HCMV in AIDS patients with [90]. This synthetic oligonucleotide works by binding to the mRNA of the HCMV immediate-early (MIE) gene, preventing its translation and thereby inhibiting viral replication. However, the drug has several limitations as an antiviral and is no longer widely used. Oligonucleotide-based drugs have several challenges: delivery, off-target effects, short half-life, limited spectrum and high cost.

All the oligonucleotide-based antiviral approaches against HCMV are still at pre-clinical stage. There are several challenges that must be addressed before they can be considered for use in the clinical setting.

## 5. Conclusions

HCMV infection is a significant global health problem, especially in immunocompromised individuals and congenitally infected infants. The virus can remain latent in the body, making it challenging to eliminate and increasing the risk of reoccurrence. Currently, there is no licensed vaccine against human cytomegalovirus (HCMV). Despite ongoing research and development efforts, developing a safe and effective vaccine for HCMV has been challenging due to the virus’s complex biology and ability to evade the immune system. This renders the role of antivirals critical in the management of HCMV infection. Unfortunately, the efficacy of licensed antiviral drugs against HCMV is also limited due to side effects, cross-resistance and other factors. It is challenging to develop a safe and effective antiviral against HCMV. In such times, using oligonucleotides to inhibit viral replication could be a desirable strategy. A major advantage of oligonucleotides is that they are designed rationally and specifically bind to target nucleic acid sequences or proteins. Recent advancements in bioinformatics and genome sequence technologies improved the in silico assessment of off-target interaction [91]. Such bioinformatics tools can predict the potential off-target effects by comparing the sequence of the antisense oligonucleotide sequence to the target mRNA and identifying any mismatches with greater precision and resolution [92]. Several reports suggest that RNAi, Ribozyme, CRISPR/Cas9 and aptamers are among the therapies that show good results in both in-vitro and in-vivo studies [68,76,81,89]. Such targeted therapy has shown antiviral activity with lower toxicity and may prove to be a promising milestone in developing therapeutic strategies for HCMV infection soon. Some limitations to this approach include the mode of delivery, stability and immunogenicity. The scientific community is dedicated to finding a solution for these limitations. Several delivery vehicles have been reported in recent years, such as lipid nanoparticle [93], polymer delivery vehicle [94] or oligos conjugated to the targeting ligand [95], which show effective results for eliminating various viral replication. Although these approaches are still in the early stage of development, more research and clinical evaluation are needed to determine their efficacy and safety against HCMV infection.

## Figures and Tables

**Figure 1 viruses-15-01358-f001:**
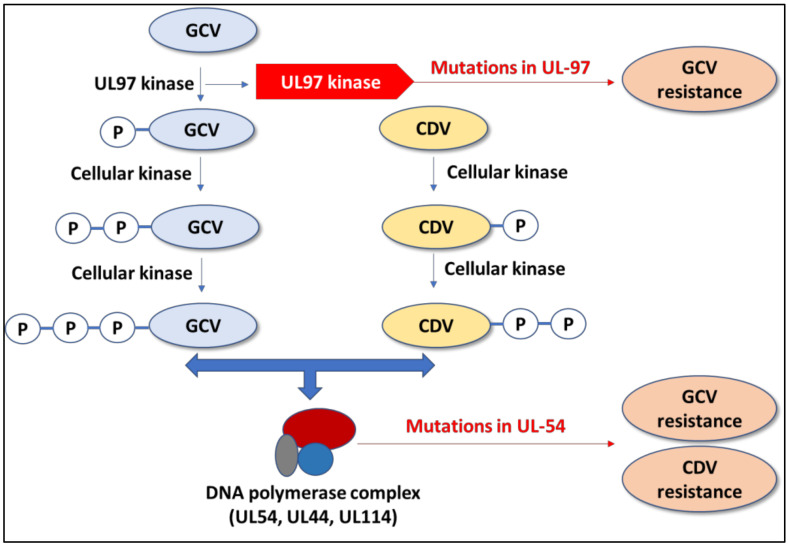
Mechanism of CMV resistance. The first phosphorylation of ganciclovir (GCV) is mediated by viral UL97 kinase. Mutation in UL-97 gene will confer resistance. DNA Polymerase is the final site of action for both GCV and Cidofovir (CDV). Mutation in UL-54 gene may confer resistance to both GCV and CDV.

**Figure 2 viruses-15-01358-f002:**
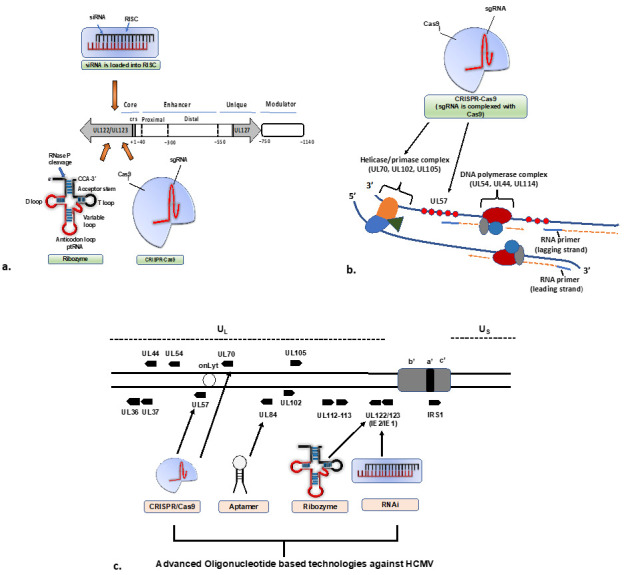
Oligonucleotide-based technologies used to target HCMV a. The HCMV major immediate early promoter (MIEP) and enhancer region is a master HCMV IE gene expression regulator. The MIEP contains a promoter region (+1 to −40 bp from the transcriptional start site of IE1/2), an enhancer region (−40 to −500 bp), (**a**) unique region (−550 to −750 bp) and a modulator (−750 to −1140 bp). siRNA, Ribozymes and CRISPR-Cas9-based technology have been used to target the IE1 and IE2 proteins, the main IE effectors encoded by UL122/UL123 ORF by alternative splicing. (**b**) HCMV replicates its genome using a viral DNA polymerase complex and a helicase-primase complex. This complex works together concertedly to synthesize both the leading and lagging strand. The helicase-primase complex comprises several subunits, including UL70, UL102 and UL105, whereas the DNA polymerase complex comprises UL54, UL44 and UL114. The UL57 is an essential gene that forms filament on ssDNA to stimulate polymerase and helicase-primase function. CRISPR-Cas9-based technology has been used to target the UL57 and UL70. (**c**) The schematic illustration shows different genes targeted by oligonucleotide-based approaches.

**Table 1 viruses-15-01358-t001:** Antiviral agents approved for treatment or prevention of HCMV infection.

Generic Name	Mechanism of Action	Route of Administration	Major Toxicity
Ganciclovir and Valganciclovir	Inhibition of viral DNA replication by targeting DNA polymerase	Intravenous, Oral	Bone marrow suppression, encephalopathy, carcinogenicity and possibly hepatotoxicity and reduced fertility
Foscarnet	Pyrophosphate analog/Inhibition of viral DNA replication by targeting DNA polymerase	Intravenous	Nephrotoxicity, hypocalcemia, electrolytes imbalance, genital ulceration
Cidofivir	Nucleotide analog/Inhibition of viral DNA replication by targeting DNA polymerase	Intravenous	Nephrotoxicity
Letermovir	Binds to components of the terminase complex (UL56/UL89)	Intravenous, Oral	Bone marrow suppression and nephrotoxicity
Maribavir	Inhibition of CMV UL97 gene product	Oral	Taste disturbance and gastrointestinal

## Data Availability

Primary literature reviewed is available on request.

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
