# Peer review of "An Update on Current Antiviral Strategies to Combat Human Cytomegalovirus Infection"

_viruses, 2023, doi:10.3390/v15061358_

Round 1
Reviewer 1 Report
In the manuscript entitled “An update on current antiviral strategies to combat Human cytomegalovirus infection”, authors Panda, Parashar and Viswanathan provide commentary on the state of antiviral therapies against HCMV infections. The authors first review HCMV biology to provide background and the rationale for the development of HCMV therapeutics before describing those that are in use and others that may have potential in a clinical setting.
Major points:
1. Figures: figure legends are necessary
- Figure 1: The layer of proteins between the capsid and envelope is called the matrix or tegument, not the shell.
- Figure 2: The figure shows the organization of the HCMV genome but does not indicate why certain open reading frames were highlighted. The figure also depicts the various technologies that have been used to target CMV infections in vitro and in some cases in vivo. Do the arrows point to specific open reading frames that have been targeted? Again a figure legend would clarify some of these questions.
Minor points:
Please address the following:
Introduction
- Lines 28-29: The authors state that the genome size is approximately 235 kbp +/1.9 kb and reference a review article by Ye et al. (2020). It is sufficient to state that genome is approximately 235 kbp, or better yet to provide references for the original literature such as NM Suárez et al. (2019).
- Lines 30-33: the HCMV genome structure (isomers, unique long, unique short, inverted repeats, etc.) is not completely described.
- Lines 33-36: No reference is provided
- Line 35: Please confirm that the authors mean virion, not vesicles.
- Lines 74-77: The authors briefly discuss HCMV in the context of transplantation but do not elaborate on the phrases “troll of transplantation” or “troll of tolerance”.
- Lines 83-84: The authors state that there are no relevant antibody treatments for HCMV infection but Cytogam IGIV has been used to treat transplant patients.
Strategies to Combat HCMV Infections:
The section on antiviral drugs is lacking a discussion of resistance to these medications.
- Lines 99-103: Please clarify how ganciclovir inhibits HCMV infection.
- Lines 116-120: As written, the statements suggest that VPD and VCD were used to treat HCMV infections in the clinical setting. These drugs were only used in MCMV-infected mice.
- Lines 119-123: The potential mode of action of VPD/VCD was not clearly articulated.
- Lines 129-131: The authors barely mention mutations that confer resistance to antiviral drugs and should expand on this topic in the context of current strategies used to treat resistant or refractory infections observed in patients.
- Lines 139-144: No references are provided.
- Lines 153-156: Please clarify how letermovir inhibits HCMV infection. How does it affect the terminase complex and viral replication.
- Lines 160-161: The FDA has approved maribavir for treatment of HCMV infections and the results of Phase III trials were published by Avery et al (2022).
The approaches described in the section on RNAi-based therapeutics have been validated in vitro and in limited studies in small animal models, not in clinical settings. The section does not include a discussion of fomivirsen, an antisense oligonucleotide that was used to treat CMV retinitis in AIDS patients.
- Lines 186-187: The study cited by the authors involved rat CMV, not HCMV.
- Lines 197-201: More details about this study should be included. A superficial description of the approach and results was provided by the authors. For example, which host miRNAs?
- Lines 214-216: The authors should be more specific in describing the viral targets such as the shared exon 3 of IE1 and IE2 which are transcribed from an RNA that is alternatively spliced.
- Lines 219-222: One of the targets is assemblin, not assembling.
- Lines 222-223: This statement regarding references 38 and 39 is rather vague. One study targeted the IE gene IE2 of HCMV and the other target capsid proteins of MCMV. Please clarify.
- Lines 243-245: Please confirm that authors mean multiplex strategies, not multiple strategies.
- Lines 248-252: The authors should elaborate more on the results from this study. They mention that there was clear “depletion” when the essential genes were targeted. Was this the important result from the study?
- Lines 256-257: Was the IE region also targeted in the THP-1 cells? Please clarify and provide more context for this experiment.
- Lines 259-260: What is the reference for this study?
- Lines 260-262: Please clarify and provide more details andcontext for this experiment.
- Line 272: Please confirm that this is the correct reference.
- Lines 278-282: The in vivo studies appear to be directed at latent MCMV infection. The authors should provide more details on how this study was executed.
- Lines 287-288: The authors seem to be specifically describing nucleic acid aptamers here. Studies using peptide aptamers are discussed in the following paragraph. These are two distinct strategies and this should be clarified.
Author Response
Reviewer 1:
In the manuscript entitled “An update on current antiviral strategies to combat Human cytomegalovirus infection”, authors Panda, Parashar and Viswanathan provide commentary on the state of antiviral therapies against HCMV infections. The authors first review HCMV biology to provide background and the rationale for the development of HCMV therapeutics before describing those that are in use and others that may have potential in a clinical setting.
Major points:
- Figures: figure legends are necessary
- Figure 1: The layer of proteins between the capsid and envelope is called the matrix or tegument, not the shell.
- Figure 2: The figure shows the organization of the HCMV genome but does not indicate why certain open reading frames were highlighted. The figure also depicts the various technologies that have been used to target CMV infections in vitro and in some cases in vivo. Do the arrows point to specific open reading frames that have been targeted? Again a figure legend would clarify some of these questions.
Response: As suggested we have modified figures 1 and 2.
Minor points:
Please address the following:
Introduction
- Lines 28-29: The authors state that the genome size is approximately 235 kbp +/1.9 kb and reference a review article by Ye et al. (2020). It is sufficient to state that genome is approximately 235 kbp, or better yet to provide references for the original literature such as NM Suárez et al. (2019).
Response: As suggested we have done changes (Page no. 2, Line nos. 52, Ref 9 ).
- Lines 30-33: the HCMV genome structure (isomers, unique long, unique short, inverted repeats, etc.) is not completely described.
Response: As suggested we have described genome structure of HCMV. (Page no. 2, Line nos- 53-59)
- Lines 33-36: No reference is provided
Response: Reference added in the revised manuscript (Page no. 2, Line nos 63, Ref no. 12).
- Line 35: Please confirm that the authors mean virion, not vesicles.
Response: We have done the changes in the revised manuscript. (Page no. 2, Line nos- 61)
- Lines 74-77: The authors briefly discuss HCMV in the context of transplantation but do not elaborate on the phrases “troll of transplantation” or “troll of tolerance”.
Response: We have done the changes in the revised manuscript. (Page no. 3, Line nos- 98-102)
- Lines 83-84: The authors state that there are no relevant antibody treatments for HCMV infection but Cytogam IGIV has been used to treat transplant patients.
Response: As suggested we have corrected statement (Page no. 3, Line nos.107-120).
Strategies to Combat HCMV Infections:
The section on antiviral drugs is lacking a discussion of resistance to these medications.
Response: Discussion of resistance has been included in revised version of manuscript. (Page no. 5, 6-, Line nos. 194-195, 229-235).
- Lines 99-103: Please clarify how ganciclovir inhibits HCMV infection.
Response: Modified statement in the revised manuscript for more clarification of statement. (Page no. 4, Line nos.129-137).
- Lines 116-120: As written, the statements suggest that VPD and VCD were used to treat HCMV infections in the clinical setting. These drugs were only used in MCMV-infected mice.
Response: In the revised version, we have made a separate segment for VPD and VCD and corrected. (Page no. 5, Line nos. 211-222)
- Lines 119-123: The potential mode of action of VPD/VCD was not clearly articulated.
Response: Statement clearly articulated in the revised manuscript. (Page no. 5, Line nos. 211-222)
- Lines 129-131: The authors barely mention mutations that confer resistance to antiviral drugs and should expand on this topic in the context of current strategies used to treat resistant or refractory infections observed in patients.
Response: Discussion of resistance has been included in revised version of manuscript. (Page no. 5, 6-, Line nos. 194, 229-235).
- Lines 139-144: No references are provided.
Response: Reference added. (Page no. 4, Line nos. 177, Reference no 39 ).
- Lines 153-156: Please clarify how letermovir inhibits HCMV infection. How does it affect the terminase complex and viral replication.
Response: We have clarified and explained statement in the revised manuscript (Page no. 5, Line nos. 186-193).
- Lines 160-161: The FDA has approved maribavir for treatment of HCMV infections and the results of Phase III trials were published by Avery et al (2022).
Response: In the revised version, we have updated the information (Page no. 5, Line nos. 207-209).ost-Transplant: Results From a Phase 3 Randomized Clinical Trial
The approaches described in the section on RNAi-based therapeutics have been validated in vitro and in limited studies in small animal models, not in clinical settings. The section does not include a discussion of fomivirsen, an antisense oligonucleotide that was used to treat CMV retinitis in AIDS patients.
Response: In the revised version, we have updated the information (Page no. 10, Line nos. 380-387).ost-Transplant
- Lines 186-187: The study cited by the authors involved rat CMV, not HCMV.
Response: Thank you for your comment, we have removed the statement in the revised manuscript.
- Lines 197-201: More details about this study should be included. A superficial description of the approach and results was provided by the authors. For example, which host miRNAs?
Response: In the revised version, we have updated the information (Page no. 7, Line nos. 257-259).ost-Transplant
- Lines 214-216: The authors should be more specific in describing the viral targets such as the shared exon 3 of IE1 and IE2 which are transcribed from an RNA that is alternatively spliced.
Response: Statement have been included in the revised version of manuscript (Page no. 7-8; Line No 274-282).
- Lines 219-222: One of the targets is assemblin, not assembling.
Response: Correction done (Page no. 8, line no. 286)
- Lines 222-223: This statement regarding references 38 and 39 is rather vague. One study targeted the IE gene IE2 of HCMV and the other target capsid proteins of MCMV. Please clarify.
Response: In the revised version, we have removed the reference with MCMV. (Page no.8 Line nos. 290).
- Lines 243-245: Please confirm that authors mean multiplex strategies, not multiple strategies.
Response: Typo mistake corrected in the revised version of manuscript (Page No. 8 Line no- 312).
- Lines 248-252: The authors should elaborate more on the results from this study. They mention that there was clear “depletion” when the essential genes were targeted. Was this the important result from the study?
Response: Thank you for your suggestion. We have elaborated the result in this section (Page No. 8, Line no 316-322).
- Lines 256-257: Was the IE region also targeted in the THP-1 cells? Please clarify and provide more context for this experiment.
Response: Thank you for your suggestion. We have elaborated the result in this section (Page No. 8-9, Line no 323-327).
- Lines 259-260: What is the reference for this study?
Response: Thank you for comment, we have removed the statement and reference.
- Lines 260-262: Please clarify and provide more details and context for this experiment.
Response: Thank you for comment, we have removed the statement and reference.
- Line 272: Please confirm that this is the correct reference.
Response: We have corrected the reference in the revised manuscript (Reference no. 71).
- Lines 278-282: The in vivo studies appear to be directed at latent MCMV infection. The authors should provide more details on how this study was executed.
Response: In the revised version, we have removed the statement.
- Lines 287-288: The authors seem to be specifically describing nucleic acid aptamers here. Studies using peptide aptamers are discussed in the following paragraph. These are two distinct strategies and this should be clarified.
Response: Thank you for your comment, we have updated the information (Page no. 9, Line nos. 344-349).ost
Reviewer 2 Report
The review by Panda et al. aims at summarizing the status of current and prospective antiviral strategies against HCMV. After listing the FDA-approved therapy options, they provide numerous nucleic-acid based therapy options that might be used in the future as soon as the problem with their targeted delivery will be solved. I think this review is very interesting but requires major revisions before it can be published in viruses.
I offer my point of view below for consideration by the authors.
Major comments/questions:
My main criticism on this review is that the authors should pay much more attention on referring to the primary literature but not own publications (i.e. Ref. 29). Similarly, reference 7 (and many others) are not suitable in their respective context. Often references were omitted, making it impossible for the reader discriminating whether the statement is based on previous findings or just reflects the opinion of the authors. The authors should point out that the only FDA-approved drugs are describeded in (i) all other (ii)-(vi) depict experimental in vitro strategies that work in the lab but none of them (to my knowledge) was tested in animal and/or clinical phase I or phase II studies. If so, they need to indicate at which developmental state these novel therapeutic options are.
Both figures must be improved and figure legends should be provided.
Minor comments/questions:
Lines 16-18: “due to its unique life cycle, mutations, and latency. As HCMV is a herpes virus, it demonstrates a unique pathogenesis by establishing a lifelong persistence in the host through a chronic and latent state of infection.” - This is somewhat redundant and not unique (other Herpesviruses are also capable to establish lifelong persistence with similar pathogeneses). Moreover, it is mainly the chronic but not the latent state that is associated with pathogenesis. The authors may wish to make this clearer.
Line 27: “belongs to the genus of Herpesvirus and the betaherpervirinae subfamily” - this was the case in former ICTV-classifications- however in nowadays HCMV is classified in the genus CYTOMEGALOVIRUS - see https://www.ncbi.nlm.nih.gov/pmc/articles/PMC2995426/ https://ictv.global/report/chapter/herpesviridae/herpesviridae
Line 29: “encodes a complex functional protein” – it encodes numerous functional proteins.
Line 30: “ has an E- type genomic structure consisting of two large reverse domains called Long (L) and short (S).” - reference(s) should be given for “E-type genomic structure” and “reverse domains”. This reviewer believes the genomic regions are named “unique long (UL)” and “unique short (US)”.
Line 34: “tegument” - this is how it is commonly named - but not “shell” as indicated in Fig.1.
Lines 44-48 - This reviewer suggests to remove this as the second sentence to line 28.
Figure 1- For my personal point of view this is one of the worst schematics of HCMV (e.g. “shell”?, or “glycoprotein” despite the fact that HCMV encodes various different glycoprotein complexes). I don´t think this figure helps the reader for the understanding of this particular review, which concentrates on strategies to combat HCMV infection. I think it might be better to show for example the entire HCMV replication cycle highlighting the potential targets of the approved and future intervention strategies. The authors may wish to ask the journal for the permission for a reprint of a beautiful schematic of the HCMV-particle or its replication cycle.
Lines 60-65: This reviewer suggests to remove this as the first sentence to line 27 (i.e. first all herpesviruses, before it gets specific about HCMV).
Linees 66 and 69: “during natural infection, within a host, it can infect a number of cell types” is redundant to “The virus shows a wide range of cellular tropism”.
Line 81: “The gastrointestinal tract, liver, and central nervous system are the primary site for symptomatic infection.” Should be removed to line 71.
Line 83f: “In addition, there are no relevant vaccines or immunoglobulins available to combat the infection” - This is not true ! Polyclonal HCMV hyperimmunoglobulins (HIG) are safe, well tolerated and effective at controlling virus multipication after SOT (Santhanakrishnan K. et al., 2019; Alsuliman T. et al., 2018; Bonaros N. et al., 2008). Kagan et al, 2019 show that biweekly administration of 200 IU/kg HIG efficiently prevents maternal-fetal transmission in the first trimester. Co-administration of two monoclonal antibodies (anti-gH (MSL-109) + anti-UL131a) reduced the post-transplant incidence of HCMV infection, delayed viremia, & reduced HCMV disease after KTR (Ishida JH et al., 2016). Even though out of the scope of their review, the authors should at least mention the recent advances made in the field.
Line 94: “antiviral drugs for HCMV act mainly as inhibitors of viral DNA polymerase” - attention Letermovir does not target the DNA polymerase but the terminase complex.
Importantly, the authors should also refer to the pUL97 kinase inhibitor Maribavir/LIVTENCITY that is FDA-approved for the Treatment of Adults With Post-transplant Cytomegalovirus (CMV) Infection And/or Disease That Are Refractory (With or Without Resistance) to One or More Prior Therapies (https://pubmed.ncbi.nlm.nih.gov/35147913/).
Lines 99ff: “Ganciclovir inhibits viral DNA replication by converting it into ganciclovir-5'-triphosphate (ganciclovir-TP). Ganciclovir converts into triphosphate by” - Should be rephrased to make clear that Ganciclovir is converted by kinases. The authors may wish to emphasize that HCMV-encoded pUL97 phosphorylates GCV (https://pubmed.ncbi.nlm.nih.gov/10428917/).
The authors should include the fact that VAL, GCV are used for prevention and therapy of congenital CMV https://www.mdpi.com/1422-0067/20/6/1388 https://link.springer.com/article/10.1007/s00430-017-0512-3
Line 113ff: Valpromide (VPD) and Valnoctamide (VCD) - may be listed but this reviewer strongly recommends to move it to the end (to keep the DNA-pol inhibitors VAL, GCV and CDV together) but more importantly to indicate that these are to my knowledge no FDA-approved drugs for the treatment/prevention of HCMV-disease.
Line 131: “in the HCMV phosphotransferase that confers resistance” - please indicate the name of the HCMV-encoded phosphotransferase (i.e. pUL97).
Line 142: “site of DNA polymerase”- make clear that it is the viral DNA-pol pUL54 and not the cellular DNA-pol.
Line 153: “HCMV terminase complex” - consider to indicate the target protein (i.e. pUL56).
Line 161: see comment above - maribavir is meanwhile approved by the FDA.
Table 1: compare side effects with https://www.ncbi.nlm.nih.gov/pmc/articles/PMC7284540/ ….
For GCV - toxicity - add Neutropenia, Thrombocytopenia, Neurotoxicity and exchange “Hematological” for “hematotoxic”For FOS - toxicity - add Nephrotoxicity, Hypocalcemia, Electrolytes imbalance, Genital ulceration
For LMV - remove „Inhibit CMV replication by“ as this is also absent for all other inhibitors.
Line174: “presence of double-stranded RNA (dsRNA) similar to the target gene's sequence” is redundant to line 182 “specifically targets and degrade mRNA molecules that are complementary to the sequence of siRNA”
Line 185: “21-23nt synthetic RNA” - make clear whether “siRNA” or “miRNA”
Lines 172-201: ii. RNAi-based therapeutics against CMV: the authors mention studies that target UL54 or UL122, 123 but none of them is indicated in Fig. 2.
Figure 2 The figure legend is missing. It is not clear to this reviewer, why some ORFs are in the same and others are in different colors!? Why are some pieces of the RNA in red and others in blue? It remains unclear, which of the indicated strategies targets which of the highlighted ORFs.
Line 207: “delivery of ribozymes to infected cells through direct injection” - it remains elusive to this reviewer how this targeted delivery into infected cells should be achieved.
Line 212: “ribozyme-based strategies have been 212 considered a vital gene-silencing tool for viral infection.” - please provide references
Line 223 “also targeted the early genes of HCMV” - indicate which genes were targeted!
Lines 245 + 251: The authors explain that UL122/123 and UL57 or UL70 were successfully targeted by CRISPR/Cas9 - but again none of those is highlighted in Fig. 2!?
Lines 267ff: For TALEN-based therapeutics the authors switched from HCMV to MCMV. Are there no studies to HUMAN CMV? As this is the title and scope of their review, the authors should focus on HCMV and only include animal studies if required for development of novel anti-HCMV therapeutics. In addition, they mention that “TALEN plasmid transfected before the virus infection inhibits MCMV growth in cell culture by cleaving specific MCMV M80/80.5 sequences” - this is for my point of understanding PROPHYLAXIS but not THERAPY and I cannot imagine how such a strategy would be translated to the human population, which needs to be convinced getting transduced with TALENS prior to potential infection with HCMV. What are the HCMV- homologs of the targeted M80 and M80.5 and why are they not indicated in Fig. 2?
Lines 291ff: “Based on previous reports, aptamers have already been designed to inhibit several functional proteins involved in viral entry into the host cell, such as a viral envelope or viral receptor-binding proteins.” - The authors should refer to these original publications that show targeting of HCMV-encoded envelope or receptor proteins blocked HCMV replication.
Lines 322ff: “Several reports suggest that RNAi, Ribozyme, CRISPR/Cas9, TALEN, and aptamer are among the therapies that show impressive results in both in-vitro and in-vivo studies.” - see comment above - the authors should point out which of the mentioned strategies is at which stage.
Lines 328ff: “Several delivery vehicles have been reported in recent years, such as lipid nanoparticle, polymer delivery vehicle, or oligos conjugated to the targeting ligand, which show effective results for eliminating various viral replication.” - as this is a review - the authors should refer to the corresponding literature.
Line 334f: “Data acquisition and interpretation” - I don´t think any Data were acquired for this review.
Typos:
Line 19 “Immunocompromised” - Immunocompromized.
Line 22: “infection Therefore” - add a point after “infection”
Line 44: “beta-herpesvirus” - change for betaherpesvirus as in line 28.
Line 96: „Ganciclovir(GCV)“ - add space.
Line179: “miRNA exert gene” - seems like an excess space in front of “exert”
Line 192: Xiaofei et al[32] - missing dot after “al.”
Lines 220+222: “capsid scaffolding protein (CSP) and the assembling.” - remove the “g” from “assembling”
Line 258: “important factor which control cell growth” - change for “controls”
Fig 2: The figure title needs to be corrected: “Oligonucleotide based technology used to target” - change for “technologies”.
Line 341: "The authors declare no conflict of interest“ - remove the first “.
Author Response
Reviewer 2
Comments and Suggestions for Authors
The review by Panda et al. aims at summarizing the status of current and prospective antiviral strategies against HCMV. After listing the FDA-approved therapy options, they provide numerous nucleic-acid based therapy options that might be used in the future as soon as the problem with their targeted delivery will be solved. I think this review is very interesting but requires major revisions before it can be published in viruses.
I offer my point of view below for consideration by the authors.
Major comments/questions:
My main criticism on this review is that the authors should pay much more attention on referring to the primary literature but not own publications (i.e. Ref. 29). Similarly, reference 7 (and many others) are not suitable in their respective context. Often references were omitted, making it impossible for the reader discriminating whether the statement is based on previous findings or just reflects the opinion of the authors. The authors should point out that the only FDA-approved drugs are describeded in (i) all other (ii)-(vi) depict experimental in vitro strategies that work in the lab but none of them (to my knowledge) was tested in animal and/or clinical phase I or phase II studies. If so, they need to indicate at which developmental state these novel therapeutic options are.
Both figures must be improved and figure legends should be provided.
Response: As suggested, we have removed our own publication. In addition, we have clarified all the information. In addition, we have improved the figure and legends.
Minor comments/questions:
Lines 16-18: “due to its unique life cycle, mutations, and latency. As HCMV is a herpes virus, it demonstrates a unique pathogenesis by establishing a lifelong persistence in the host through a chronic and latent state of infection.” - This is somewhat redundant and not unique (other Herpesviruses are also capable to establish lifelong persistence with similar pathogeneses). Moreover, it is mainly the chronic but not the latent state that is associated with pathogenesis. The authors may wish to make this clearer.
Response: We agree with the reviewer and have now edited this statement (Page no 1; Line no. 16, 17 and 18)
Line 27: “belongs to the genus of Herpesvirus and the betaherpervirinae subfamily” - this was the case in former ICTV-classifications- however in nowadays HCMV is classified in the genus CYTOMEGALOVIRUS - see https://www.ncbi.nlm.nih.gov/pmc/articles/PMC2995426/ https://ictv.global/report/chapter/herpesviridae/herpesviridae
Response: We have done corrections as per the suggestion (Page no 2; Line no. 51)
Line 29: “encodes a complex functional protein” – it encodes numerous functional proteins.
Response: Corrections done (Page no 2; Line no. 52)
Line 30: “has an E- type genomic structure consisting of two large reverse domains called Long (L) and short (S).” - reference(s) should be given for “E-type genomic structure” and “reverse domains”. This reviewer believes the genomic regions are named “unique long (UL)” and “unique short (US)”.
Response: We have modified the statement in the revised manuscript (Page no. 2; Line no. 53-59)
Line 34: “tegument” - this is how it is commonly named - but not “shell” as indicated in Fig.1.
Response: We have included modified figure in the revised manuscript.
Lines 44-48 - This reviewer suggests to remove this as the second sentence to line 28.
Response: As suggested we have removed statement in the revised manuscript .
Figure 1- For my personal point of view this is one of the worst schematics of HCMV (e.g. “shell”?, or “glycoprotein” despite the fact that HCMV encodes various different glycoprotein complexes). I don´t think this figure helps the reader for the understanding of this particular review, which concentrates on strategies to combat HCMV infection. I think it might be better to show for example the entire HCMV replication cycle highlighting the potential targets of the approved and future intervention strategies. The authors may wish to ask the journal for the permission for a reprint of a beautiful schematic of the HCMV-particle or its replication cycle.
Response: Thank you for pointing out. We have modified the figure in the revised version of manuscript.
Lines 60-65: This reviewer suggests to remove this as the first sentence to line 27 (i.e. first all herpesviruses, before it gets specific about HCMV).
Response: The suggestion is well taken and the sentence has been modified accordingly (Page no 3; Line no. 85)
Lines 66 and 69: “during natural infection, within a host, it can infect a number of cell types” is redundant to “The virus shows a wide range of cellular tropism”.
Response: Thank you for your suggestion. We have changed the statement in the revised version of manuscript (Page no. 3; Line Nos. 90).
Line 81: “The gastrointestinal tract, liver, and central nervous system are the primary site for symptomatic infection.” Should be removed to line 71.
Response: Thank you for your suggestion. We have removed statement in the revised manuscript..
Line 83f: “In addition, there are no relevant vaccines or immunoglobulins available to combat the infection” - This is not true ! Polyclonal HCMV hyperimmunoglobulins (HIG) are safe, well tolerated and effective at controlling virus multipication after SOT (Santhanakrishnan K. et al., 2019; Alsuliman T. et al., 2018; Bonaros N. et al., 2008). Kagan et al, 2019 show that biweekly administration of 200 IU/kg HIG efficiently prevents maternal-fetal transmission in the first trimester. Co-administration of two monoclonal antibodies (anti-gH (MSL-109) + anti-UL131a) reduced the post-transplant incidence of HCMV infection, delayed viremia, & reduced HCMV disease after KTR (Ishida JH et al., 2016). Even though out of the scope of their review, the authors should at least mention the recent advances made in the field.
Response: Thank you for the suggestion. We have modified the statements in the revised version of manuscript. As suggested we have included statements with references (Page No.3 Line Nos. 107-120).
Line 94: “antiviral drugs for HCMV act mainly as inhibitors of viral DNA polymerase” - attention Letermovir does not target the DNA polymerase but the terminase complex.
Response: We have removed statement in the revised manuscript.
Importantly, the authors should also refer to the pUL97 kinase inhibitor Maribavir/LIVTENCITY that is FDA-approved for the Treatment of Adults With Post-transplant Cytomegalovirus (CMV) Infection And/or Disease That Are Refractory (With or Without Resistance) to One or More Prior Therapies (https://pubmed.ncbi.nlm.nih.gov/35147913/).
Response: We have now included this detail in the revised manuscript. (Page no. 5, Line no. 207-209)
Lines 99ff: “Ganciclovir inhibits viral DNA replication by converting it into ganciclovir-5'-triphosphate (ganciclovir-TP). Ganciclovir converts into triphosphate by” - Should be rephrased to make clear that Ganciclovir is converted by kinases. The authors may wish to emphasize that HCMV-encoded pUL97 phosphorylates GCV (https://pubmed.ncbi.nlm.nih.gov/10428917/).
Response: The point is well noted and has now been included in the revised manuscript. (Page no. 4, Line no. 129-132)
The authors should include the fact that VAL, GCV are used for prevention and therapy of congenital CMV https://www.mdpi.com/1422-0067/20/6/1388 https://link.springer.com/article/10.1007/s00430-017-0512-3
Response: We thank the authors for the valuable suggestion and have updated the same in the manuscript. (Page no. 4, Line no. 139-144, Ref no. 33)
Line 113ff: Valpromide (VPD) and Valnoctamide (VCD) - may be listed but this reviewer strongly recommends to move it to the end (to keep the DNA-pol inhibitors VAL, GCV and CDV together) but more importantly to indicate that these are to my knowledge no FDA-approved drugs for the treatment/prevention of HCMV-disease.
Response: As suggested we have placed Valpromide (VPD) and Valnoctamide (VCD) with mentioned statement (Page no. 5; Line nos 211-222).
Line 131: “in the HCMV phosphotransferase that confers resistance” - please indicate the name of the HCMV-encoded phosphotransferase (i.e. pUL97).
Response: We have modified the statement in the revised manuscript (Page no. 4 ; Line nos 157).
Line 142: “site of DNA polymerase”- make clear that it is the viral DNA-pol pUL54 and not the cellular DNA-pol.
Response: We have done changes in the revised manuscript (Page no. 4 Line nos 175).
Line 153: “HCMV terminase complex” - consider to indicate the target protein (i.e. pUL56).
Response: We have done correction in the revised manuscript (Page No. 5, Line Nos. 189-190).
Line 161: see comment above - maribavir is meanwhile approved by the FDA.
Response: Thank you for suggestion. We have incorporated maribavir separately (Page No. 5 , Line Nos. 197-209).
Table 1: compare side effects with https://www.ncbi.nlm.nih.gov/pmc/articles/PMC7284540/ ….
For GCV - toxicity - add Neutropenia, Thrombocytopenia, Neurotoxicity and exchange “Hematological” for “hematotoxic”
For FOS - toxicity - add Nephrotoxicity, Hypocalcemia, Electrolytes imbalance, Genital ulceration
For LMV - remove „Inhibit CMV replication by“as this is also absent for all other inhibitors.
Response: We have done changes in table as per suggestions.
Line174: “presence of double-stranded RNA (dsRNA) similar to the target gene's sequence” is redundant to line 182 “specifically targets and degrade mRNA molecules that are complementary to the sequence of siRNA”
Response: Thank you for your suggestion. We have removed the statement in the revised version.
Line 185: “21-23nt synthetic RNA” - make clear whether “siRNA” or “miRNA”
Response: We have modified the statement. (Page no. 7; Line Nos. 250-251).
Lines 172-201: ii. RNAi-based therapeutics against CMV: the authors mention studies that target UL54 or UL122, 123 but none of them is indicated in Fig. 2.
Response: As suggested in the revised version, we have modified the fig and legends.
Figure 2 The figure legend is missing. It is not clear to this reviewer, why some ORFs are in the same and others are in different colors!? Why are some pieces of the RNA in red and others in blue? It remains unclear, which of the indicated strategies targets which of the highlighted ORFs.
Response: In the revised version, we have modified the fig and legends.
Line 207: “delivery of ribozymes to infected cells through direct injection” - it remains elusive to this reviewer how this targeted delivery into infected cells should be achieved.
Response: In the revised version, we have modified the statement. (Page no. 7; Line nos- 267-268)
Line 212: “ribozyme-based strategies have been 212 considered a vital gene-silencing tool for viral infection.” - please provide references
Response: Reference added (Page no. 7. Line nos 274, Reference no 61 ).
Line 223 “also targeted the early genes of HCMV” - indicate which genes were targeted!
Response: We have added genes details in the revised manuscript (Page No. 7-8, Line Nos. 272-299).
Lines 245 + 251: The authors explain that UL122/123 and UL57 or UL70 were successfully targeted by CRISPR/Cas9 - but again none of those is highlighted in Fig. 2!?
Response: As suggested modified Figure 2 incorporated in the revised manuscript.
Lines 267ff: For TALEN-based therapeutics the authors switched from HCMV to MCMV. Are there no studies to HUMAN CMV? As this is the title and scope of their review, the authors should focus on HCMV and only include animal studies if required for development of novel anti-HCMV therapeutics. In addition, they mention that “TALEN plasmid transfected before the virus infection inhibits MCMV growth in cell culture by cleaving specific MCMV M80/80.5 sequences” - this is for my point of understanding PROPHYLAXIS but not THERAPY and I cannot imagine how such a strategy would be translated to the human population, which needs to be convinced getting transduced with TALENS prior to potential infection with HCMV. What are the HCMV- homologs of the targeted M80 and M80.5 and why are they not indicated in Fig. 2?
Response: We thank the reviewer for the critical input and have now removed the section on TALENs .
Lines 291ff: “Based on previous reports, aptamers have already been designed to inhibit several functional proteins involved in viral entry into the host cell, such as a viral envelope or viral receptor-binding proteins.” - The authors should refer to these original publications that show targeting of HCMV-encoded envelope or receptor proteins blocked HCMV replication.
Response: In the revised version, we have modified the statement (Page No.9 , Line Nos.346-349).
Lines 322ff: “Several reports suggest that RNAi, Ribozyme, CRISPR/Cas9, TALEN, and aptamer are among the therapies that show impressive results in both in-vitro and in-vivo studies.” - see comment above - the authors should point out which of the mentioned strategies is at which stage.
Response: In the revised version, we have modified the sentence and mentioned the stage in each strategies paragraph.
Lines 328ff: “Several delivery vehicles have been reported in recent years, such as lipid nanoparticle, polymer delivery vehicle, or oligos conjugated to the targeting ligand, which show effective results for eliminating various viral replication.” - as this is a review - the authors should refer to the corresponding literature.
Response: Added (Page No 11, Line Nos. 440-443. Reference no. 80-82).
Line 334f: “Data acquisition and interpretation” - I don´t think any Data were acquired for this review.
Response: We have removed this statement.
Typos:
Line 19 “Immunocompromised” - Immunocompromized.
Response: Correction done. (Page no. 1, line nos. 19)
Line 22: “infection Therefore” - add a point after “infection”
Response: Done. (Page no. 1, line nos. 22)
Line 44: “beta-herpesvirus” - change for betaherpesvirus as in line 28.
Response: Done. (Page no. 3, line nos. 88)
Line 96: „Ganciclovir(GCV)“ - add space.
Response: Done. (Page no. 3, line nos. 125)
Line179: “miRNA exert gene” - seems like an excess space in front of “exert”
Response: Correction done in the revised version. (Page no. 7, line nos. 244)
Line 192: Xiaofei et al[32] - missing dot after “al.”
Response: Correction done in the revised version. (Page no. 7, line nos. 252)
Lines 220+222: “capsid scaffolding protein (CSP) and the assembling.” - remove the “g” from “assembling”
Response: Correction done in the revised version (Page no. 8, line no. 286)
Line 258: “important factor which control cell growth” - change for “controls”
Response: In the revised version, we have removed this line.
Fig 2: The figure title needs to be corrected: “Oligonucleotide based technology used to target” - change for “technologies”.
Response: Figure legend modified in the revised version.
Line 341: "The authors declare no conflict of interest“ - remove the first “.
Response: In the revised version, we have removed.
Reviewer 3 Report
Thank-you for asking me to review the article entitled “An update on current antiviral strategies to combat Human cytomegalovirus infection” by Panda et al. The review article focuses mainly on antiviral strategies, both new and old, to combat HCMV with a focus on the immunocompromised host. The discussion on new technologies is interesting but the review of currently used antiviral agents in clinical practice is incomplete.
Minor Comments:
-line 22 punctuation error. Requires a full stop after “infection” prior to starting next sentence
-line 74 would change AIDS to advanced HIV as certain clinical HCMV phenotypes are AIDS defining illnesses in those with advanced HIV
-line 76 grammatical error “outcomes’”
-the description of HCMV now as the “troll of tolerance” requires further expansion as opposed to simply a reference to this concept
-line 84: need to define what is meant by “relevant immunoglobulin” as CMV hyperimmunoglobulin is an available treatment modality
-line 85: the authors mention critically unwell patients (with only reference to the elderly) although they may also consider mentioning illness in premature neonates as post-natal CMV infection in the NICU is a common occurrence, presenting as viral sepsis or “culture negative sepsis” and may require antiviral treatment in critically unwell neonates
-would recommend brief description of CMV vaccines in the development pipeline
-line 85: error in sentence grammar and syntax which makes the author’s point confusing
-line 102 change “cell” to “cells”
-although central venous catheters are often already present in those requiring ganciclovir (ie. immunocompromised hosts” I would not say that this is the main reason to insert a CVL nor the most commonly discussed draw back/complication of the drug
-without having to read reference 19 in full it is confusing why weight gain is considered a positive outcome to CMV infection as described by the authors
-line 133-134: grammatical errors makes the sentence confusing to the reader. What is meant by the term “delayed by cidofovir”
-line 139: although it is true that foscarnet is commonly used in ganciclovir resistance, it is even more commonly used in cases of early haematopoeitic stem cell transplant where it has benefits over ganciclovir in that it is less marrow suppressive. The authors should also indicate that foscarnet is only available as an IV formulation (this is included in the table but the authors should be consistent with their discussion of each agent and it’s formulation in the text ie. either include in text of each agent or include only in the table)
-the authors may also wish to briefly include that both ganciclovir and foscarnet can be used as an intravitreal preparation (for CMV retinitis) given the repeated mention of patients with AIDS defining CMV illnesses
-line 146-147: syntax and grammar error makes the sentence confusing to read
-line 148-149 tense error: change “have been” to “was”
-I would avoid including brand names next to generic names as these may be different in different parts of the world
-Table 1: need to be consistent with terminology ie. “renal” v “nephrotoxicity” and “bone marrow” v “haematological” and need to define what is meant by “neurotoxicity”
-line 199-201: the sentence is repetitive
-line 210-231: sentences require grammatical restructuring
-line 222 & 228: clarify “viral growth”, is this by viral culture?
-line 234-237: sentence structure and grammatical errors
-line 255: sentence tense error
-in the text relating to each mechanism of action of the new antiviral technologies please reference the follow-up figure so that it cues the reader to associate the text with the figure. It would be helpful to associate each box in the figure with a letter so that in the text, for each new technology the figure reference could be for example Figure 2(a), (b), (c) etc. This would more easily draw the reader to the specific part of the figure addressed in the paragraph
-need consistency in the in text referencing format
-line 294-295: sentence structure needs correcting “specific target molecule specifically”
-line 288-291: repetitive sentences
-line 291-293 requires reference
-line 321-322: can the authors be clearer in what is meant in this sentence?
Major comments:
-I would recommend that in the introduction the authors discuss the burden of CMV in the population both in terms of severe disease (such as that discussed in the immunocompromised host with end organ disease) but also the population prevalence of CMV (ie. seropositivity prevalence in populations of interest), those that have asymptomatic disease and are seropositive, as well as the less severe disease caused by CMV in the immunocompetent host such as rash, fever, pharyngitis, lymphadenopathy etc. in order to give the reader the true representation of the spectrum of HCMV disease/infection
-although no current HCMV vaccine exists, there are many in the pipeline. In the body of the “strategies to combat HCMV infection” section, the manuscript would benefit from a brief discussion on the issues related to vaccine development for HCMV and new (and old) vaccine technology that may be able to address these issues, or on the contrary why a vaccine strategy may not be the best approach. This is addressed very superficially in the conclusion which seems like an unusual place to do so
-line 108: the statement “CNS damage will not be reversed” needs further elaboration. As it is true that structural damage from in utero CMV infection (eg. Polymicrogyria) is not reversible with treatment, CNS disease in itself is an indication for treatment of symptomatic congenital CMV in order to improve developmental outcomes, hearing, and ocular disease. The authors do not discuss the clinical outcome rationale behind treatment in congenital CMV and the sentence almost makes it seem like there is no benefit to treatment
-the authors need to expand on what is meant by “haematologic parameters”
-line 126: the statement that cidofovir is the most widely used antiviral in HCMV disease in this population is no longer accurate, unless the authors are suggesting this on a historic basis and if this is the case then a reference should be provided
-the paragraph describing cidofovir needs to include a sentence on nephrotoxicity as opposed to just being included in Table 1 as it is particularly clinically relevant
-can the authors include a figure explaining with visual representation of the mechanisms of the various currently available antiviral drugs and where mutations in the various pathways would result in resistance? This would also make the text description of mechanisms of action and resistance more clear to the reader. Cross resistance is mentioned in the conclusion and this may be helpful in addressing this more clearly as well
-authors need to elaborate in what clinical context letermovir would be used as prophylaxis in HIV infection
-line 155: what do the authors mean by this statement?
-line 145: there needs to be significant expansion ie. a whole paragraph dedicated to maribavir. Studies on this new antiviral agent in refractory disease or in circumstances when other agents are contraindicated have now been reported in the broader literature. The statement as it currently reads is out of date and makes the review both less clinically useful and less comprehensive.
-As it pertains to brincidofovir, although it has anecdotal clinical utility over cidofovir, particularly given its more positive renal profile, phase 3 trials have not shown improved clinical outcomes and at present it is very difficult to access the drug given its very limited manufacturing
-the authors should include a few short sentences on the potential benefit of the addition of leflunomide to antiviral treatment in specific circumstances given it’s properties. This is done in many transplant centres in refractory cases and thus should be addressed
-the authors should address recent studies on the use of high dose valaciclovir as an antiviral agent, shown to have benefit in prevention of congenital CMV in mothers with early pregnancy infections
-a comprehensive review should also, at least briefly, discuss the role of anti-CMV cytotoxic lymphocyes (CTLs) in HSCT patients
-line 304 and 308 contradict each other as it pertains to the authors comments on “stability” of the molecule
-the conclusion overall is repetitive and I would recommend considerable restructuring
Author Response
Reviewer 3
Comments and Suggestions for Authors
Thank-you for asking me to review the article entitled “An update on current antiviral strategies to combat Human cytomegalovirus infection” by Panda et al. The review article focuses mainly on antiviral strategies, both new and old, to combat HCMV with a focus on the immunocompromised host. The discussion on new technologies is interesting but the review of currently used antiviral agents in clinical practice is incomplete.
Response: Thank you for the comment. We have modified the antiviral section in the revised version (Page no. 4, 5, 6; line no. 130-149, 165-167, 185-187, 190-191, 195-196, 198-229)
Minor Comments:
-line 22 punctuation error. Requires a full stop after “infection” prior to starting next sentence
Response: Correction done. Page nos. 1 Line nos. 22
-line 74 would change AIDS to advanced HIV as certain clinical HCMV phenotypes are AIDS defining illnesses in those with advanced HIV
Response: Correction done. (Page nos. 3 Line nos. 96)
-line 76 grammatical error “outcomes’”
Response: Correction done. Page nos. 3; Line nos. 100
-the description of HCMV now as the “troll of tolerance” requires further expansion as opposed to simply a reference to this concept
Response: As suggested we have added statement (Page No. 3, Line Nos. 98 -102).
-line 84: need to define what is meant by “relevant immunoglobulin” as CMV hyperimmunoglobulin is an available treatment modality
Response: As suggested we have added explanation for relevant immunoglobulin (Page No. 3, Line Nos. 107-120 ).
-line 85: the authors mention critically unwell patients (with only reference to the elderly) although they may also consider mentioning illness in premature neonates as post-natal CMV infection in the NICU is a common occurrence, presenting as viral sepsis or “culture negative sepsis” and may require antiviral treatment in critically unwell neonates
Response: As suggested we have modified the statement in the revised manuscript (Page no. 3; Line nos 103-106).
-would recommend brief description of CMV vaccines in the development pipeline
Response: As suggested we have incorporated brief description of CMV vaccine (Page No. 11-12, Line Nos. 392-406).
-line 85: error in sentence grammar and syntax which makes the author’s point confusing
Response: We have modified the sentence in the revised version of manuscript (Page No. 4, Line Nos. 117-120).
-line 102 change “cell” to “cells”
Response: We have removed the sentence in modified version.
-although central venous catheters are often already present in those requiring ganciclovir (ie. immunocompromised hosts” I would not say that this is the main reason to insert a CVL nor the most commonly discussed draw back/complication of the drug
Response: we have modified the sentence. (Page nos-4, Line nos-139)
-without having to read reference 19 in full it is confusing why weight gain is considered a positive outcome to CMV infection as described by the authors
Response: We have done changes in the revised manuscript (Page No. 5, Line Nos. 214-215).
-line 133-134: grammatical errors makes the sentence confusing to the reader. What is meant by the term “delayed by cidofovir”
Response: Statement removed in the revised manuscript (Page No. 5, Line Nos. 159).
-line 139: although it is true that foscarnet is commonly used in ganciclovir resistance, it is even more commonly used in cases of early haematopoeitic stem cell transplant where it has benefits over ganciclovir in that it is less marrow suppressive. The authors should also indicate that foscarnet is only available as an IV formulation (this is included in the table but the authors should be consistent with their discussion of each agent and it’s formulation in the text ie. either include in text of each agent or include only in the table)
Response: The route of administration of the approved drugs has now been included in the text (Page no., 4 line no. 174)
-the authors may also wish to briefly include that both ganciclovir and foscarnet can be used as an intravitreal preparation (for CMV retinitis) given the repeated mention of patients with AIDS defining CMV illnesses
Response: The point has now been included. (Page no. 4, Line nos. 133)
-line 146-147: syntax and grammar error makes the sentence confusing to read
Response: We have modified sentence for better understanding (Page No.5, Line Nos.179 ).
-line 148-149 tense error: change “have been” to “was”
Response: Correction done. (Page No.5 , Line Nos. 183 ).
-I would avoid including brand names next to generic names as these may be different in different parts of the world
Response: The brand names have now been removed. (Table 1)
-Table 1: need to be consistent with terminology ie. “renal” v “nephrotoxicity” and “bone marrow” v “haematological” and need to define what is meant by “neurotoxicity”
Response: Modified table included in the revised manuscript. Neurotoxicity replaced with specific side effect of encephalopathy
-line 199-201: the sentence is repetitive
Response: Repeat statement removed from revised manuscript.
-line 210-231: sentences require grammatical restructuring
Response: As suggested we have modified sentences. (Page No.7 , Line Nos. 272-299).
-line 222 & 228: clarify “viral growth”, is this by viral culture?
Response: Correction done (Page no. 8 ; Line nos 287, 294).
-line 234-237: sentence structure and grammatical errors
Response: Correction Done (Page No. 8 , Line Nos. 302-306).
-line 255: sentence tense error
Response: Correction Done (Page No. 8-9, Line Nos. 323-327).
-in the text relating to each mechanism of action of the new antiviral technologies please reference the follow-up figure so that it cues the reader to associate the text with the figure. It would be helpful to associate each box in the figure with a letter so that in the text, for each new technology the figure reference could be for example Figure 2(a), (b), (c) etc. This would more easily draw the reader to the specific part of the figure addressed in the paragraph
Response: In the revised version, we have done the correction accordingly.
-need consistency in the in text referencing format
Response: In the revised version, we have done the correction accordingly.
-line 294-295: sentence structure needs correcting “specific target molecule specifically”
Response: Correction done in the revised version (Page No. 9, Line Nos. 344-345).
-line 288-291: repetitive sentences
Response: Corrections made in the revised version (Page No. 9, Line Nos. 346-347).
-line 291-293 requires reference
Response: Added references in the revised manuscript (Page no. 9, Line no. 348, Reference nos. 73, 74 ).
-line 321-322: can the authors be clearer in what is meant in this sentence?
Response: We have discussed in detail in the revised version of manuscript (Page No. 11, Line Nos. 431-433).
Major comments:
-I would recommend that in the introduction the authors discuss the burden of CMV in the population both in terms of severe disease (such as that discussed in the immunocompromised host with end organ disease) but also the population prevalence of CMV (ie. seropositivity prevalence in populations of interest), those that have asymptomatic disease and are seropositive, as well as the less severe disease caused by CMV in the immunocompetent host such as rash, fever, pharyngitis, lymphadenopathy etc. in order to give the reader the true representation of the spectrum of HCMV disease/infection.
Response: Thank you so much for pointing out. In the revised version, we have added accordingly. (Page no. 1-2 , line no. 28-49)
-although no current HCMV vaccine exists, there are many as suggested in the pipeline. In the body of the “strategies to combat HCMV infection” section, the manuscript would benefit from a brief discussion on the issues related to vaccine development for HCMV and new (and old) vaccine technology that may be able to address these issues, or on the contrary why a vaccine strategy may not be the best approach. This is addressed very superficially in the conclusion which seems like an unusual place to do so
Response: Thank you for your comment. As the current MS is on current antiviral strategies so we have not incorporated vaccine strategies, now we have added vaccine strategies in brief (Page No. 10-11 , Line Nos. 392-417).
-line 108: the statement “CNS damage will not be reversed” needs further elaboration. As it is true that structural damage from in utero CMV infection (eg. Polymicrogyria) is not reversible with treatment, CNS disease in itself is an indication for treatment of symptomatic congenital CMV in order to improve developmental outcomes, hearing, and ocular disease. The authors do not discuss the clinical outcome rationale behind treatment in congenital CMV and the sentence almost makes it seem like there is no benefit to treatment
Response: We thank the reviewer for pointing this out and we have now elaborated in the revised version. (Page no. 4, Line nos. 134-143)
-the authors need to expand on what is meant by “haematologic parameters”
Response: In the revised version, we have modified the term. (Page no. 4, Line no. 146-147)
-line 126: the statement that cidofovir is the most widely used antiviral in HCMV disease in this population is no longer accurate, unless the authors are suggesting this on a historic basis and if this is the case then a reference should be provided
Response: We thank the reviewer for pointing this out and have now removed the statement.
-the paragraph describing cidofovir needs to include a sentence on nephrotoxicity as opposed to just being included in Table 1 as it is particularly clinically relevant
Response: The details have now been included. (Page no. 4, line no. 163-167)
-can the authors include a figure explaining with visual representation of the mechanisms of the various currently available antiviral drugs and where mutations in the various pathways would result in resistance? This would also make the text description of mechanisms of action and resistance more clear to the reader. Cross resistance is mentioned in the conclusion and this may be helpful in addressing this more clearly as well
Response: Thank you for your suggestion. We have incorporated modified figure in the revised manuscript.
-authors need to elaborate in what clinical context letermovir would be used as prophylaxis in HIV infection
Response: We have elaborated statement in the revised manuscript (Page No. 5, Line Nos. 183-186 ).
-line 155: what do the authors mean by this statement?
Response: Thank you for your comment, we have removed statement in the revised manuscript.
-line 145: there needs to be significant expansion ie. a whole paragraph dedicated to maribavir. Studies on this new antiviral agent in refractory disease or in circumstances when other agents are contraindicated have now been reported in the broader literature. The statement as it currently reads is out of date and makes the review both less clinically useful and less comprehensive.
Response: We have added para on maribavir in the revised manuscript (Page No. 5, Line Nos. 197-209).
-As it pertains to brincidofovir, although it has anecdotal clinical utility over cidofovir, particularly given its more positive renal profile, phase 3 trials have not shown improved clinical outcomes and at present it is very difficult to access the drug given its very limited manufacturing
Response: We have removed statement in the revised manuscript.
-the authors should include a few short sentences on the potential benefit of the addition of leflunomide to antiviral treatment in specific circumstances given it’s properties. This is done in many transplant centres in refractory cases and thus should be addressed
Response: Based on suggestion we have incorporated the details. (Page no. 6, Line no. 223-228)
-the authors should address recent studies on the use of high dose valaciclovir as an antiviral agent, shown to have benefit in prevention of congenital CMV in mothers with early pregnancy infections
Response: Thank you for your comment and now we have added mentioned studies in the revised manuscript. (Page no. 4, line no. 134-138)
-a comprehensive review should also, at least briefly, discuss the role of anti-CMV cytotoxic lymphocyes (CTLs) in HSCT patients
Response: As suggested we have now incorporated the role of anti-CMV cytotoxic lymphocyes (CTLs) in HSCT patients.
-line 304 and 308 contradict each other as it pertains to the authors comments on “stability” of the molecule
Response: As suggested we have done correction.
-the conclusion overall is repetitive and I would recommend considerable restructuring
Response: Thank you for your suggestion. We have modified conclusion.
Reviewer 4 Report
Summary:
The review by Panda et al. summarizes the current status of antiviral strategies to counteract HCMV infection. This includes a description of the currently available antiviral drugs, which have already been approved for HCMV treatment. It details the mechanism of action of these antiviral compounds and highlights the urgent need for alternative, less toxic treatment options. In this regard, the review provides insights into in-vitro and in-vivo studies using different nucleic acid-based approaches to target HCMV replication that may promote the development of novel therapeutics with distinct mode of actions and better safety profiles in the future.
General concept comments:
With antiviral strategies to combat HCMV infection, Panda et al. summarize a topic, which is up to date and of major significance as long as no effective vaccine against HCMV is available. The strength of this review is that besides already licensed anti-HCMV drugs, the authors summed up research on a number of nucleic acid-based approaches used to inhibit HCMV infection that may have the potential to open up the path for the development of novel intervention options in the clinics. However the reviewer offers following suggestions for the authors’ consideration: (i) Some parts of the manuscript are difficult to read, and would benefit from an editing of English language. (ii) Several references don’t seem to be appropriate or are incorrect and need revision, e.g. [5], [6], [7], [9], [29], [46]. Also some references are missing (see specific comments). (iii) Certain sections of the manuscript could be presented in a more structured manner, e.g. the introduction switches from a specific HCMV description to a general herpesvirus introduction and back to HCMV again. For chapter ii. RNA-based therapeutics against HCMV, the general description of the mode of action of siRNAs or miRNAs should be finalized before switching to the use of this technology as antiviral strategy to combat viral infections (iv) Generally, the figures could be improved by providing corresponding figure legends.
Specific comments:
Line 16: should read “Human cytomegalovirus”
Line 17: should read “herpesvirus”
Line 22: punctuation is missing
Line 28: should read “betaherpesvirinae”
Lines 29-30: HCMV does not encode “a complex functional protein” but numerous functional proteins” or “a complex set of functional proteins”
Lines 30-33: regarding the description of the HCMV genome organization, the reviewer is only aware of designations like internal repeats/terminal repeats = IR/TR, and unique short/unique long = US/UL
Line 44: should read “betaherpesvirus” or “beta-herpesvirus”
Figure 1: the labeling requires editing as letters are overlapping and the term shell should be replaced by the term tegument
Line 62: should read “human herpesvirus 1-8”
Line 64: should read “ betaherpesviruses” or beta-herpesviruses”
Line 69: should read “neutrophils, and hepatocytes”
Line 99: the term “by converting it into” is misleading and should be replaced by “is converted into”
Lines 108-109: In congenital infection, existing CNS damage will not be reversed by antivirals. à please provide reference
Line 113: Since VPD and VCD are not explicitly approved for the treatment of HCMV disease by FDA or EMA, it is recommended to list them after the approved drugs to make this point clear.
Lines 122-123: what is meant by “or both the virus and the cell simultaneously interact to inhibit the attachment”. This sentence needs further clarification.
Line 125: should read “cidofovir was the first”
The reference to table 1 is missing in the text.
Line 161: Of note: Maribavir was recently approved in the USA for the treatment of post-transplant CMV infection/disease that is refractory to treatment (with or without genotypic resistance) with ganciclovir, valganciclovir, cidofovir or foscarnet in adults and paediatric (≥ 12 years of age and weighing ≥ 35 kg) patients (https://pubmed.ncbi.nlm.nih.gov/35147913/).
Line 179: Space between miRNA and exert has to be removed.
Line 179: The reference [29] seems to belong to the previous sentence: “The RNAi-based therapeutic belongs to a wave of advanced…. – if so, please correct.
Line 184: self-citation [29] not appropriate at this point, as the general mode of action of miRNAs is described!!
Line 198: should read “has been investigated”
Line 212: “now days” should read “nowadays”
Line 220: what is meant by “assembling” in this context. Does the M1GS RNA target an HCMV protein required for viral particle assembly? Please clarify.
Line 234: should read “has been developed to treat” à leave out “and focus”
Line 235: should read “can be used”
Line 236: should read “directly cleaving DNA and RNA”
Line 251: “expressing the essential genes” has to be changed to “targeting the essential genes”
Line 259: should read “controls cell growth”
Lines 259-260: Studies confirmed the association of HCMV pUL38 protein with mTORC1 during successful infection by HCMV. à citation is missing!
Lines 260-262: A study showed that mTORC1 inhibitor suppressed HCMV replication in-vitro and recued the HCMV reactivation using CRISPR technology [46]. Sentence and citation is not correct. In the indicated publication [46], no mTORC1 inhibitors were used to reduce the incidence of HCMV reactivation. These findings are based on publications that are not related to the usage of CRISPR/Cas9 technology: https://pubmed.ncbi.nlm.nih.gov/23269449/, https://pubmed.ncbi.nlm.nih.gov/25446337/.
Instead, [46] applied the CRISPR-Cas9 system to knock out TSC2, a negative regulator of mTORC1. The authors showed that pUL38 was still able to activate mTORC1 in TSC2-deficient cells, which is required for efficient HCMV replication. Thus, no CRISPR/Cas9-mediated block of HCMV replication is described in this publication.
Line 272: [45] no TALENs described in this publication, please add correct reference.
Line 291-293: Based on previous reports, aptamers have already been designed to inhibit several function proteins involved in viral entry into the host cell, such as a viral envelope or viral receptor-binding proteins. à please provide reference for this statement
Line 297: should read “peptide aptamers from”
Figure 2 is somehow misleading as the indicated ORFs do not reflect the ORFs, which are targeted by the different nucleic acid-based approaches specified. The figure would greatly benefit from including the ORFs which are the targets of the individual antiviral strategies and the respective arrows directly pointing to them.
Figure 2: should read “different oligonucleotide based technologies”
Line 299: should read “expression library bind to”
Line 303: the word primers must be replaced by aptamers
Line 311: The reviewer would suggest to add individuals/patients/hosts/people after immunocompromised

Author Response
Reviewer 4
Comments and Suggestions for Authors
Summary:
The review by Panda et al. summarizes the current status of antiviral strategies to counteract HCMV infection. This includes a description of the currently available antiviral drugs, which have already been approved for HCMV treatment. It details the mechanism of action of these antiviral compounds and highlights the urgent need for alternative, less toxic treatment options. In this regard, the review provides insights into in-vitro and in-vivo studies using different nucleic acid-based approaches to target HCMV replication that may promote the development of novel therapeutics with distinct mode of actions and better safety profiles in the future.
General concept comments:
With antiviral strategies to combat HCMV infection, Panda et al. summarize a topic, which is up to date and of major significance as long as no effective vaccine against HCMV is available. The strength of this review is that besides already licensed anti-HCMV drugs, the authors summed up research on a number of nucleic acid-based approaches used to inhibit HCMV infection that may have the potential to open up the path for the development of novel intervention options in the clinics. However the reviewer offers following suggestions for the authors’ consideration: (i) Some parts of the manuscript are difficult to read, and would benefit from an editing of English language. (ii) Several references don’t seem to be appropriate or are incorrect and need revision, e.g. [5], [6], [7], [9], [29], [46]. Also some references are missing (see specific comments). (iii) Certain sections of the manuscript could be presented in a more structured manner, e.g. the introduction switches from a specific HCMV description to a general herpesvirus introduction and back to HCMV again. For chapter ii. RNA-based therapeutics against HCMV, the general description of the mode of action of siRNAs or miRNAs should be finalized before switching to the use of this technology as antiviral strategy to combat viral infections (iv) Generally, the figures could be improved by providing corresponding figure legends.
Response: Thank you for appreciation for addition of nucleic acid-based approaches and number of suggestions for improvement of review.
- We have tried to improve English language in the revised manuscript.
- Thank you for indicating the oversight for references and accordingly done correction in the revised version of manuscript.
- As suggested we have improved manuscript in structured manner.
- We have added corresponding figure legends based on suggestion.
Specific comments:
Line 16: should read “Human cytomegalovirus”
Response: In the revised version, we have modified (Line no – 16, pg nos- 1)
Line 17: should read “herpesvirus”
Response: In the revised version, we have modified (Line no – 17, pg nos- 1)
Line 22: punctuation is missing
Response: In the revised version, we have modified (Line no – 22, pg nos- 1)
Line 28: should read “betaherpesvirinae”
Response: In the revised version, we have modified (Line no – 51, pg nos- 2)
Lines 29-30: HCMV does not encode “a complex functional protein” but numerous functional proteins” or “a complex set of functional proteins”
Response: In the revised version, we have modified. (Line no – 52, pg nos- 2)
Lines 30-33: regarding the description of the HCMV genome organization, the reviewer is only aware of designations like internal repeats/terminal repeats = IR/TR, and unique short/unique long = US/UL
Response: We have corrected the description (Page no. 2, Line nos. 53-59)
Line 44: should read “betaherpesvirus” or “beta-herpesvirus”
Response: Done (Page No 3. , Line Nos. 88).
Figure 1: the labeling requires editing as letters are overlapping and the term shell should be replaced by the term tegument
Response: Thank you for your comment and accordingly we have modified figure in the revised version of manuscript.
Line 62: should read “human herpesvirus 1-8”
Response: Done (Page No.3 , Line Nos 86. ).
Line 64: should read “ betaherpesviruses” or beta-herpesviruses”
Response: Done (Page No 3. , Line Nos. 88).
Line 69: should read “neutrophils, and hepatocytes”
Response: In the modified version, we have removed this statement.
Line 99: the term “by converting it into” is misleading and should be replaced by “is converted into”
Response: In the modified version, we have removed this statement (Page no. 4, Line no. 153)
Lines 108-109: In congenital infection, existing CNS damage will not be reversed by antivirals. à please provide reference
Response: As suggested we have added reference in the revised manuscript (Reference no. 32).
Line 113: Since VPD and VCD are not explicitly approved for the treatment of HCMV disease by FDA or EMA, it is recommended to list them after the approved drugs to make this point clear.
Response: Thank you for your suggestion, accordingly we have shifted VPD and VCD (Page No. 5, Line Nos. 211-222).
Lines 122-123: what is meant by “or both the virus and the cell simultaneously interact to inhibit the attachment”. This sentence needs further clarification.
Response: We have modified sentence with more clarity (Page No. 5, Line Nos. 220-222).
Line 125: should read “cidofovir was the first”
Response: : In the modified version, we have corrected this (Page No. 5, Line Nos. 150).
The reference to table 1 is missing in the text.
Response: Done (Page No.3 , Line Nos. 123).
Line 161: Of note: Maribavir was recently approved in the USA for the treatment of post-transplant CMV infection/disease that is refractory to treatment (with or without genotypic resistance) with ganciclovir, valganciclovir, cidofovir or foscarnet in adults and paediatric (≥ 12 years of age and weighing ≥ 35 kg) patients (https://pubmed.ncbi.nlm.nih.gov/35147913/).
Response: Details have now been included (Page no. 5, Line nos. 207-209)
Line 179: Space between miRNA and exert has to be removed.
Response: Done (Page No. 7, Line Nos. 244).
Line 179: The reference [29] seems to belong to the previous sentence: “The RNAi-based therapeutic belongs to a wave of advanced…. – if so, please correct.
Response: Done correction. (Page no. 7, line no. 245, Ref no. 53)
Line 184: self-citation [29] not appropriate at this point, as the general mode of action of miRNAs is described!!
Response: Thank you for your comment and accordingly done correction.
Line 198: should read “has been investigated”
Response: In the revised version, we have modified the statement (Line no – 258, pg nos- 7)
Line 212: “now days” should read “nowadays”
Response: In the revised version, we have modified (Line no – 273, pg nos- 7)
Line 220: what is meant by “assembling” in this context. Does the M1GS RNA target an HCMV protein required for viral particle assembly? Please clarify.
Response: In the revised version, we have modified the statement (Line no –283-286 pg nos- 8)
Line 234: should read “has been developed to treat” à leave out “and focus”
Response: In the revised version, we have modified the statement (Line no – 302, pg nos- 8)
Line 235: should read “can be used”
Response: In the revised version, we have modified the statement (Line no – 303, pg nos- 8)
Line 236: should read “directly cleaving DNA and RNA”
Response: In the revised version, we have modified the statement (Line no – 303-304, pg nos- 8)
Line 251: “expressing the essential genes” has to be changed to “targeting the essential genes”
Response: In the revised version, we have modified the statement (Line no – 318-319, pg nos- 8)
Line 259: should read “controls cell growth”
Response: In the revised version, we have removed the statement.
Lines 259-260: Studies confirmed the association of HCMV pUL38 protein with mTORC1 during successful infection by HCMV. à citation is missing!
Response: In the revised version, we have removed the statement.
Lines 260-262: A study showed that mTORC1 inhibitor suppressed HCMV replication in-vitro and recued the HCMV reactivation using CRISPR technology [46]. Sentence and citation is not correct. In the indicated publication [46], no mTORC1 inhibitors were used to reduce the incidence of HCMV reactivation. These findings are based on publications that are not related to the usage of CRISPR/Cas9 technology: https://pubmed.ncbi.nlm.nih.gov/23269449/, https://pubmed.ncbi.nlm.nih.gov/25446337/.
Response: In the revised version, we have removed this study and reference.
Instead, [46] applied the CRISPR-Cas9 system to knock out TSC2, a negative regulator of mTORC1. The authors showed that pUL38 was still able to activate mTORC1 in TSC2-deficient cells, which is required for efficient HCMV replication. Thus, no CRISPR/Cas9-mediated block of HCMV replication is described in this publication.
Response: Thank you for your comment, In the revised version, we have removed this study and reference.
Line 272: [45] no TALENs described in this publication, please add correct reference.
Response: Thank you for pointing out. We have corrected the reference. In addition, We have now removed TALEN in detail as suggested by majority of reviewers. (Page no. 9, Line no. 338, Ref no. 71)
Line 291-293: Based on previous reports, aptamers have already been designed to inhibit several function proteins involved in viral entry into the host cell, such as a viral envelope or viral receptor-binding proteins. à please provide reference for this statement
Response: We have added the reference in revised manuscript. (Ref no. 73, 74).
Line 297: should read “peptide aptamers from”
Response: Thank you for suggestions. We have changed in the revised version of manuscript (Line nos- 369, pg no. 10)
Figure 2 is somehow misleading as the indicated ORFs do not reflect the ORFs, which are targeted by the different nucleic acid-based approaches specified. The figure would greatly benefit from including the ORFs which are the targets of the individual antiviral strategies and the respective arrows directly pointing to them.
Response: Thank you for suggestions, accordingly we have modified fig.
Figure 2: should read “different oligonucleotide-based technologies”
Response: We have done changes in the revised manuscript. (Line nos- 350, pg no. 10)
Line 299: should read “expression library bind to”
Response: We have done changes in the revised manuscript. (Line nos- 369, pg no. 10)
Line 303: the word primers must be replaced by aptamers
Response: We have done changes in the revised manuscript. (Line nos- 374, pg no. 10)
Line 311: The reviewer would suggest to add individuals/patients/hosts/people after immunocompromised
Response: As suggested we have done correction in the revised version of the manuscript. (Line nos- 420, pg no. 11)
Round 2
Reviewer 1 Report
The revised manuscript is much improved but additional changes are necessary.
Please address the following reviewer queries (line numbers correspond to marked up version):
Line 17-19: I believe the authors want to state: “As HCMV is a herpesvirus, it establishes a lifelong, persistent infection.” It is unclear what is meant by redundant pathogenesis. Please clarify.
Line 79: Please provide references for work demonstrating that the trimer plays a role in fusion of viral and cellular membranes during viral egress. Do the authors mean viral egress from endosomes or viral egress/release from cells? Please specify and provide references.
Lines 96-97: Do the authors mean “Virions contain a proteinaceous matrix called tegument that surrounds the icosahedral capsid containing the double-stranded DNA genome.” ? Please correct the figure legend and add references.
Lines 157-159: Please clarify if more than one patient was presented in the referenced study (a patient versus organ transplant recipients).
Lines 160-161: Viral DNA replication is not “mediated” by GCV. Please clarify this statement.
Lines 237-240: Please correct this sentence. Terminase is important for cleavage and packaging of viral DNA into capsids, which occurs in the nucleus after the viral DNA is replicated in the nucleus. No transport is involved.
Lines 244-246: What is meant by this statement? How does binding of letermovir to UL56 reduce “cross-resistance” with antivirals that target UL54 (viral DNA polymerase)?
Lines 253-254: This statement is incorrect. As stated above, viral DNA replication and encapsidation occur in the nucleus. The mature envelope is subsequently acquired in the cytoplasm.
Lines 254-258: Please provide a reference(s). Although UL97 has many functions, this reviewer is unaware of evidence showing that UL97 is involved in capsid assembly.
Lines 273-275: Please clarify this sentence. Do the authors mean that the drugs may interfere with attachment by interacting with the virion and/or host cell HSPG to prevent binding of virus to the cell surface?
Line 279-280: Do the authors mean that leflunomide has been used effectively to prevent CMV reactivation from latency or to treat HCMV infection that has already been reactivated? This is an important distinction. Please clarify this statement.
Lines 289-293: This sentence has lost some of its meaning due to editing. Please correct it.
Line 295: Although UL97 can modify antivirals similar to the alphaherpesvirus thymidine kinases, I do not believe it is a thymidine kinase. It has been classified as a phosphotransferase and protein kinase. Please provide references stating that UL97 has been classified as a thymidine kinase or correct this sentence.
TABLE 1: Please confer with the editor/journal whether references are required in the table. Please correct the information for letermovir—the terminase complex is UL56/UL89.
Lines 336-338: What is meant by exogenous and endogenous delivery methods? The authors changed the wording from the original submission. Please clarify.
Lines 390-392: Please clarify what is meant by depletion? Depletion of what?
Lines 392-394: This sentence is taken almost verbatim from the abstract of the referenced article. The experiments are done entirely in vitro and do not demonstrate any therapeutic effects in a clinical setting. Moreover, the experiments described did not address HCMV latency/reactivation. Please clarify this statement.
Lines 434-437: Did the authors mean “Aptamers have been previously been used to target proteins required for virus entry into the host cells, such as cellular receptors and viral envelope proteins.”?
Figure 2 and legend: The text on this figure is too small to read in some parts. Why did the authors include so much information in the legend that was not but could have been written into the text (for example parts A and B of figure)? No references were included. Part C of the figure appears to be a summary of studies reviewed in the text. Perhaps the authors could re-work the figure and focus on Part C.
Lines 479-509: This section was not in the original submission. A review of CMV vaccine strategies would be quite lengthy and may be beyond the scope of this review.
Minor corrections (typos, punctuation, formatting, references):
Line 28: ….a ubiquitous virus either existing….
Line 29: …causing severe disease in immunocompromised…..
Line 30: ….infected newborns…
Line 34: …in asymptomatic infection or in ….
Line 49: The “HCMV” abbreviation has already been defined.
Line 57: The HCMV genome is arranged in a class E structure, with two unique regions…..
Line 67: There are 162 capsomeres in the capsid….
Line 119: ….postnatal…
Lines 166-169: Please provide a reference.
Lines 179: ….and thrombocytopenia. (period was missing)
Line 203: HCMV-encoded phosphotransferase
Lines 210-214: Please provide references.
Line 216-220: Please edit this sentence and correct typos.
Line 222: viral DNA polymerase (please remove hyphen)
Lines 237-244: Please provide references.
Line 251: significant not significance
Lines 252-253: Please provide reference
Line 266-267: Please define the abbreviations VPD and VCD.
Line 271: unreported
Lines 272-273: …or free virions. (please remove comma)
Lines 314-315: “A few studies have reported the inhibition of HCMV by both siRNA and miRNA [54].” This sentence seem out of place in the paragraph and could be moved to line 320 after “(Figure 2).” Also, please confirm this is the correct reference.
Line 320: Early not earlier studies….
Line 341: Is this supposed to be a new paragraph? The formatting has changed from the original submission. Also, please delete the word majorly.
Line 342: Please delete the word functionalized.
Line 343-344: early not earlier
Lines 345-346: ….subunit from RNAse P of Escherichia coli to target the shared exon 3 of the major immediate early mRNAs (UL122-123) of HCMV [62].
Line 354: ….capsid scaffolding protein (CSP) and assemblin.
Lines 359-360: …reported that ribonuclease P-associated external guide sequences.
Line 374: Please select used or proposed.
Line 381: Two CRISPR/Cas9 strategies have been used to target….
Line 389: ….technology targeting essential viral…..
Lines 453-454: In 2009, aptamer-based gene silencing technology was tested against HCMV for the first time (Figure 2) [75].
Line 455: The study used peptide aptamer technology to interfere with….
Line 498: Did the authors mean viral vector vaccines?
Lines 523-526: Please include references.
Line 526: Such bioinformatic tools can predict…
Lines 528-530: Please provide references here.
Author Response
Reviewer 1
Open Review
Please address the following reviewer queries (line numbers correspond to marked up version):
Line 17-19: I believe the authors want to state: “As HCMV is a herpesvirus, it establishes a lifelong, persistent infection.” It is unclear what is meant by redundant pathogenesis. Please clarify.
Reply: - We have modified the statement in revised version of manuscript. (Line no. 17, Page no. 1)
Line 79: Please provide references for work demonstrating that the trimer plays a role in fusion of viral and cellular membranes during viral egress. Do the authors mean viral egress from endosomes or viral egress/release from cells? Please specify and provide references.
Reply: - Thank you for pointing out. We have removed the statement in revised version of manuscript.
Lines 96-97: Do the authors mean “Virions contain a proteinaceous matrix called tegument that surrounds the icosahedral capsid containing the double-stranded DNA genome.” ? Please correct the figure legend and add references.
Reply: - We have made changes in the revised version of manuscript. Line no. 89-90, page no. 3, Ref 22
Lines 157-159: Please clarify if more than one patient was presented in the referenced study (a patient versus organ transplant recipients).
Reply: - We have made changes in the revised version of manuscript. Line no. 124-125, page no. 4
Lines 160-161: Viral DNA replication is not “mediated” by GCV. Please clarify this statement.
Reply: - We have modified the statement in revised version of manuscript. Line no. 125-129, page no. 4
Lines 237-240: Please correct this sentence. Terminase is important for cleavage and packaging of viral DNA into capsids, which occurs in the nucleus after the viral DNA is replicated in the nucleus. No transport is involved.
Reply: - We have made changes in the revised version of manuscript. Line no. 183-184, page no. 5
Lines 244-246: What is meant by this statement? How does binding of letermovir to UL56 reduce “cross-resistance” with antivirals that target UL54 (viral DNA polymerase)?
Reply: - Thank you for pointing out. We have removed the statement in revised version of manuscript.
Lines 253-254: This statement is incorrect. As stated above, viral DNA replication and encapsidation occur in the nucleus. The mature envelope is subsequently acquired in the cytoplasm.
Reply: - We have removed the statement in revised version of manuscript.
Lines 254-258: Please provide a reference(s). Although UL97 has many functions, this reviewer is unaware of evidence showing that UL97 is involved in capsid assembly.
Reply: - We have added reference in the revised version of manuscript. Line no. 196, Ref. 51, page no. 5
Lines 273-275: Please clarify this sentence. Do the authors mean that the drugs may interfere with attachment by interacting with the virion and/or host cell HSPG to prevent binding of virus to the cell surface?
Reply: - We have removed the statement in revised version of manuscript.
Line 279-280: Do the authors mean that leflunomide has been used effectively to prevent CMV reactivation from latency or to treat HCMV infection that has already been reactivated? This is an important distinction. Please clarify this statement.
Reply: - The statement has now been clarified in the revised manuscript. Line 218-220. Pg 6
Lines 289-293: This sentence has lost some of its meaning due to editing. Please correct it.
Reply: - We have corrected the statement in revised version of manuscript. Line no. 225-227, page no. 6
Line 295: Although UL97 can modify antivirals similar to the alpha herpesvirus thymidine kinases, I do not believe it is a thymidine kinase. It has been classified as a phosphotransferase and protein kinase. Please provide references stating that UL97 has been classified as a thymidine kinase or correct this sentence.
Reply: - We have modified the statement in the revised version of manuscript. Line no. 229, page no. 6
TABLE 1: Please confer with the editor/journal whether references are required in the table. Please correct the information for letermovir—the terminase complex is UL56/UL89.
Reply: - We have corrected the information in revised version of manuscript. Table 1, page 7
Lines 336-338: What is meant by exogenous and endogenous delivery methods? The authors changed the wording from the original submission. Please clarify.
Reply: - We have clarified the statement in revised version of manuscript. Line no. 289-291, page no. 8
Lines 390-392: Please clarify what is meant by depletion? Depletion of what?
Reply: - We have modified the statement in revised version of manuscript. Line no. 338, page no. 10.
Lines 392-394: This sentence is taken almost verbatim from the abstract of the referenced article. The experiments are done entirely in vitro and do not demonstrate any therapeutic effects in a clinical setting. Moreover, the experiments described did not address HCMV latency/reactivation. Please clarify this statement.
Reply: - We have modified the statement in revised version of manuscript. Line no. 340-342, page no. 10
Lines 434-437: Did the authors mean “Aptamers have previously been used to target proteins required for virus entry into the host cells, such as cellular receptors and viral envelope proteins.”?
Reply: - We have modified the statement in revised version of manuscript. Line no. 379, page no. 11.
Figure 2 and legend: The text on this figure is too small to read in some parts. Why did the authors include so much information in the legend that was not but could have been written into the text (for example parts A and B of figure)? No references were included. Part C of the figure appears to be a summary of studies reviewed in the text. Perhaps the authors could re-work the figure and focus on Part C.
Reply: - We have modified the figure in the revised version of manuscript. Fig.2, Page no. 10.
Lines 479-509: This section was not in the original submission. A review of CMV vaccine strategies would be quite lengthy and may be beyond the scope of this review.
Reply: - We have added this portion as per the suggestion of reviewer 3.
Minor corrections (typos, punctuation, formatting, references):
Line 28: ….a ubiquitous virus either existing….
Reply: Done. Line no. 27, pg no. 1
Line 29: …causing severe disease in immunocompromised…..
Reply: Done. Line no. 28, pg no. 1
Line 30: ….infected newborns…
Reply: Done. Line no. 29, pg no. 1
Line 34: …in asymptomatic infection or in ….
Reply: Line no. 40, pg no. 1
Line 49: The “HCMV” abbreviation has already been defined.
Reply: Done. Line no. 55, pg no. 2
Line 57: The HCMV genome is arranged in a class E structure, with two unique regions…..
Reply: Done. Line no. 66, pg no. 2
Line 67: There are 162 capsomeres in the capsid….
Reply: Done. Line no. 72, pg no. 3
Line 119: ….postnatal…
Reply: We have removed the statement.
Lines 166-169: Please provide a reference.
Reply: Reference added. Line no. 132, pg no. 4, Ref no. 33
Lines 179: ….and thrombocytopenia. (period was missing)
Reply: Done. Line no. 139, pg no. 4
Line 203: HCMV-encoded phosphotransferase
Reply: Done. Line no. 153, pg no. 4
Lines 210-214: Please provide references.
Reply: Done. Line no. 163, pg no. 4, Ref no. 43
Line 216-220: Please edit this sentence and correct typos.
Reply: Done. Line no. 165-168, pg no. 5
Line 222: viral DNA polymerase (please remove hyphen)
Reply: Done. Line no. 171, page no. 5
Lines 237-244: Please provide references.
Reply: Done. Ref no. 49, Line no. 189, pg no. 6
Line 251: significant not significance
Reply: Done. Line no. 193, page no. 5
Lines 252-253: Please provide reference
Reply: Done. Ref no. 51, Line no. 196, pg no. 5
Line 266-267: Please define the abbreviations VPD and VCD.
Reply: Done. Line no. 206, page no. 5
Line 271: unreported
Reply: Done. Line no. 210, Pg no. 5
Lines 272-273: …or free virions. (please remove comma)
Reply: Done. Line no. 212, page no. 6
Lines 314-315: “A few studies have reported the inhibition of HCMV by both siRNA and miRNA [54].” This sentence seem out of place in the paragraph and could be moved to line 320 after “(Figure 2).” Also, please confirm this is the correct reference.
Reply: Done. Line no. 270-271, page no. 8
Line 320: Early not earlier studies….
Reply: Done. Line no. 271, page no. 8
Line 341: Is this supposed to be a new paragraph? The formatting has changed from the original submission. Also, please delete the word majorly.
Reply: Done. Line no. 294, page no. 9
Line 342: Please delete the word functionalized.
Reply: Done. Line no. 295, page no. 9
Line 343-344: early not earlier
Reply: Done. Line no. 296, page no. 9
Lines 345-346: ….subunit from RNAse P of Escherichia coli to target the shared exon 3 of the major immediate early mRNAs (UL122-123) of HCMV [62].
Reply: Done. Line no. 297-298, page no. 9
Line 354: ….capsid scaffolding protein (CSP) and assemblin.
Reply: Done. Line no. 306-307, page no. 9
Lines 359-360: …reported that ribonuclease P-associated external guide sequences.
Reply: done. Done. Line no. 312, page no. 9
Line 374: Please select used or proposed.
Reply: Revied the statement. Done. Line no. 322-324, page no. 9
Line 381: Two CRISPR/Cas9 strategies have been used to target….
Reply: Done. Line no. 330, page no. 9
Line 389: ….technology targeting essential viral…..
Reply: Done. Line no. 337, page no. 10
Lines 453-454: In 2009, aptamer-based gene silencing technology was tested against HCMV for the first time (Figure 2) [75].
Reply: Done. Line no. 380, page no. 11
Line 455: The study used peptide aptamer technology to interfere with….
Reply: Done. Line no. 382, page no. 11
Line 498: Did the authors mean viral vector vaccines?
Reply: Yes, corrected in the revised version. Line no. 249, Page no. 8
Lines 523-526: Please include references.
Reply: Ref no. 92, Line no. 421, pg no. 12
Line 526: Such bioinformatic tools can predict…
Reply: Done. Line no. 418, pg no. 12
Lines 528-530: Please provide references here.
Reply: Ref no. 68, 76, 81, 89, line no. 422
Reviewer 2 Report
Reviewer (comments for the authors)
I had already reviewed the first version of the manuscript of Panda et al., which definitely improved due to my initial suggestions/corrections. However, this reviewer still has the impression, that the current draft requires further changes in its structure and figures, which need to be modified prior to publication in viruses. I offer my point of view below for consideration by the authors.
Major comments
My current version of the manuscript contains a duplication of Fig.1. - remove one of both.
Within Fig. 1 I find it a little bit misleading that the viral glycoproteins/-complexes are overlaid with the lipid envelope - which makes of course no sense and does not reflect reality - they are anchored within the envelope and have an ectodomain that is exposed to the outside. That´s why, I would recommend removing the outer thick grey layer. In addition, I would suggest to draw at least 5 proteins for the pentamer and as TC and PC share at least 2 components (gH/gL) one should consider to use the same (but not different) symbols for these two in both complexes.
Line 77: references [13-15] - not sure whether they are suitable! [13] refers to tegument proteins and no primary reference for gM/gN etc!?
Lines 78f “gH, gL, and gO. It is involved in viral entry into the host cell and in the fusion of the viral envelope with cellular membranes during viral egress.” - If TC is involved in egress of the virus this reviewer and the reader would need to see the reference for this.
Paragraph from lines 81-87: “The genome contains…….. completely unknown genes” - for my opinion has to be relocated to line 56.
Lines 81f: “proteins; gH, gL, UL128, UL130, and UL-131. The genome contains various gene families, including RL11, UL14, UL18, UL25, UL82, UL120, US6, US7, US12 and US22.” - For my understanding in the field of herpesvirus research it widely accepted to differentiate the genes (UL120, UL25) from the proteins they encode, by adding a “p” for protein or “pp” for phosphoprotein or “gp” for glycoprotein (pUL128, pp65, pp71, gH). This means the authors should use the terms pUL128, pUL130 etc. when talking about the respective proteins. Alternatively they can write “proteins encoded by UL128, UL130”.
Paragraph from lines 98-104: “To date, eight herpesviruses have been identified….. tropism for differentiated hematippoietic and epitalial cells.” - for my opinion has to relocated to the very beginning- i.e. line 28.
Line 96: “cells contain a proteinaceaous matrix called tegument” - Wrong! It is the “virus” which contains the tegument.
Lines 110-117 and also lines 118-122 is kind of redundant to what is said in lines 45-49. Remove or integrate.
Line 128: “there are no approved vaccines or immunoglobulins available to combat the HCMV infection” - this is contradictory to what they now added in the next sentences. However, thanks for correcting the initial omission of this important fact.
Lines 152, 153 and 263: not sure about the numbering and ordering: Is there any “ii”? If not remove “i” in line 153. Did I understand correctly that “paragraph 2” in Line 152ff is about the currently available drugs whereas “paragraph 3” in lines 306ff discusses future alternative strategies. The authors should properly order and name their headlines to avoid confusion of the reader.
Lines 163, 203, 252, 254: see comment above - UL97 versus pUL97.
Lines 240f, 242 and 245: “UL56 and UL89” - see comment above and add “p” for protein.
Lines 254ff: “The UL97 kinase enzyme plays a critical role in this process by phosphorylating proteins involved in the assembly of the viral capsid. …. Disrupts the assembly of the viral capsid” - The authors should pay attention on their assumptions. To my knowledge pUL97 has various substrates (Rb, RNA-pol-CTD, NEC-components, nuclear Lamins, Pin1, the mRNA export factor pUL69 etc,) but to my knowledge, it does not disrupt any capsid but the nuclear envelope during nuclear egress of the capsids. Please formulate more precisely and provide the correct references. In addition, I am not aware whether the effect of Maribavir has been evaluated for all of the pUL97-substrates.
In my current version of the manuscript, Fig.2 was accidentally relocated below the Figure legend. Many of my initial comments are now included in the figure, however it is still very blurry and hardly to read. A simple reorganization of the subfigures would allow showing everything larger and better readable (e.g. move the genome in Fig. 2a to the left (one could also remove modulator and UL127) this would alow moving 2B to the left). In Fig. 2a and 2b the same icons were used for RISC, CRISPR and Ribotzyme, but however not in Fig. 2c!?! Please use the same icons consistently throughout the same figure. It might also be sufficient only showing Fig.2c with the small icons of RNA-technology from 2a as there is currently a lot of redundancy and inconsistency within the current version of the Fig2.
Legend of Fig.2 - see comment above and clearly discriminate genes from proteins!
Line 432 “XNA” - this is an uncommon acronym and should be explained to the reader.
Line 450: ”long (U L ) and unique short (U S ) sequences bound by terminal repeats” - remove space between UL, US and exchange “bound” for “flanked”. Importantly, however, the terminal repeats are not shown in the Fig2c?!? Please remove this sentence or show the TRS in the figure.
Line 492: “vaccine that stimulates T cells and neutralizes antibodies.” - Hopefully a vaccine would not neutralize antibodies but induce (neutralizing) antibodies that confer protection from disease.
Line 504: “is the identification of new viral glycoproteins”- probably there will not be discovered many new glycoproteins - but novel potent epitopes/antigens - as the authors state correctly in their next sentence.
Minor comments/questions:
Typos:
Line 18 “a chronic a” - remove second “a”.
Line 28: “ither” - “either”
Line 30: “new-borns” - “newborns”
Line 81: “UL-131” - change for “UL131”
Line 217 “in in early” - remove one “in”
Line 264: “Recently, A few” - lower letter “a”
Line 340: remove the large interruption to line 341
Line 374: “used proposed” - remove either of them
Line 540: “DP;; First” - remove the needless semicolon
Author Response
Reviewer 2
Reviewer (comments for the authors)
I had already reviewed the first version of the manuscript of Panda et al., which definitely improved due to my initial suggestions/corrections. However, this reviewer still has the impression, that the current draft requires further changes in its structure and figures, which need to be modified prior to publication in viruses. I offer my point of view below for consideration by the authors.
Major comments
My current version of the manuscript contains a duplication of Fig.1. - remove one of both.
Response: In the revised version, we have removed the duplicate figure.
Line 77: references [13-15] - not sure whether they are suitable! [13] refers to tegument proteins and no primary reference for gM/gN etc!?
Response: As suggested we have modified references. Ref. 18-21, Line no. 78, Pg no. 2
Lines 78f “gH, gL, and gO. It is involved in viral entry into the host cell and in the fusion of the viral envelope with cellular membranes during viral egress.” - If TC is involved in egress of the virus this reviewer and the reader would need to see the reference for this.
Response: We have modified the statement in revised version of manuscript. Line no. 80, Pg no. 2
Paragraph from lines 81-87: “The genome contains…….. completely unknown genes” - for my opinion has to be relocated to line 56.
Response: As suggested sentence relocated. Line no. 59-65. Page no. 2
Lines 81f: “proteins; gH, gL, UL128, UL130, and UL-131. The genome contains various gene families, including RL11, UL14, UL18, UL25, UL82, UL120, US6, US7, US12 and US22.” - For my understanding in the field of herpesvirus research it widely accepted to differentiate the genes (UL120, UL25) from the proteins they encode, by adding a “p” for protein or “pp” for phosphoprotein or “gp” for glycoprotein (pUL128, pp65, pp71, gH). This means the authors should use the terms pUL128, pUL130 etc. when talking about the respective proteins. Alternatively they can write “proteins encoded by UL128, UL130”.
Response: We have modified terminology as suggested by reviewer. Changes done in- line no. 127, 185, 186, Pg no. 4-5
Paragraph from lines 98-104: “To date, eight herpesviruses have been identified….. tropism for differentiated hematippoietic and epitalial cells.” - for my opinion has to relocated to the very beginning- i.e. line 28.
Response: As suggested sentence relocated. Line no. 29- 36. Pg no. 1
Line 96: “cells contain a proteinaceaous matrix called tegument” - Wrong! It is the “virus” which contains the tegument.
Response: Thank you for your comment, we have modified the statement. Line no. 88, Pg no. 3
Lines 110-117 and also lines 118-122 is kind of redundant to what is said in lines 45-49. Remove or integrate.
Line 128: “there are no approved vaccines or immunoglobulins available to combat the HCMV infection” - this is contradictory to what they now added in the next sentences. However, thanks for correcting the initial omission of this important fact.
Response: We have modified the statement in revised version. Line no. 106-107, pg no. 3
Lines 152, 153 and 263: not sure about the numbering and ordering: Is there any “ii”? If not remove “i” in line 153. Did I understand correctly that “paragraph 2” in Line 152ff is about the currently available drugs whereas “paragraph 3” in lines 306ff discusses future alternative strategies. The authors should properly order and name their headlines to avoid confusion of the reader.
Response: As suggested we have removed “i”. We have properly named the subheadings for better understanding in the revised manuscript .
Lines 163, 203, 252, 254: see comment above - UL97 versus pUL97.
Response: We have modified terminology as suggested by the reviewer. Line no. 127, pg no. 4
Lines 240f, 242 and 245: “UL56 and UL89” - see comment above and add “p” for protein.
Response: We have modified terminology as suggested by reviewer. Line no. 185, 186, pg no. 5
Lines 254ff: “The UL97 kinase enzyme plays a critical role in this process by phosphorylating proteins involved in the assembly of the viral capsid. …. Disrupts the assembly of the viral capsid” - The authors should pay attention on their assumptions. To my knowledge pUL97 has various substrates (Rb, RNA-pol-CTD, NEC-components, nuclear Lamins, Pin1, the mRNA export factor pUL69 etc,) but to my knowledge, it does not disrupt any capsid but the nuclear envelope during nuclear egress of the capsids. Please formulate more precisely and provide the correct references. In addition, I am not aware whether the effect of Maribavir has been evaluated for all of the pUL97-substrates.
Response: As suggested we have stated precisely and correct reference added in the revised manuscript. Ref no. 51, line no 197
In my current version of the manuscript, Fig.2 was accidentally relocated below the Figure legend. Many of my initial comments are now included in the figure, however it is still very blurry and hardly to read. A simple reorganization of the subfigures would allow showing everything larger and better readable (e.g. move the genome in Fig. 2a to the left (one could also remove modulator and UL127) this would alow moving 2B to the left). In Fig. 2a and 2b the same icons were used for RISC, CRISPR and Ribotzyme, but however not in Fig. 2c!?! Please use the same icons consistently throughout the same figure. It might also be sufficient only showing Fig.2c with the small icons of RNA-technology from 2a as there is currently a lot of redundancy and inconsistency within the current version of the Fig2.
Response: Thank you for your suggestion and accordingly modified figure.
Legend of Fig.2 - see comment above and clearly discriminate genes from proteins!
Response: We have modified figure 2 and clearly stated genes from proteins in the revised manuscript.
Line 432 “XNA” - this is an uncommon acronym and should be explained to the reader.
Response: We have modified the statement in the revised manuscript. Line no 375 pg no. 11.
Line 450:” long (U L) and unique short (U S) sequences bound by terminal repeats” - remove space between UL, US and exchange “bound” for “flanked”. Importantly, however, the terminal repeats are not shown in the Fig2c?!? Please remove this sentence or show the TRS in the figure.
Response: As suggested we have removed TRS in the revised manuscript. Line no. 352-355, Pg no. 10
Line 492: “vaccine that stimulates T cells and neutralizes antibodies.” - Hopefully a vaccine would not neutralize antibodies but induce (neutralizing) antibodies that confer protection from disease.
Response: We have modified statement in the revised manuscript. Line no. 245, Pg no. 8
Line 504: “is the identification of new viral glycoproteins”- probably there will not be discovered many new glycoproteins - but novel potent epitopes/antigens - as the authors state correctly in their next sentence.
Response: We have corrected the statement as per suggestion of reviewer. Line no. 255, Pg no. 7
Minor comments/questions:
Typos:
Line 18 “a chronic a” - remove second “a”.
Response: Done. Line no. 18, Pg no. 1
Line 28: “ither” - “either”
Response: Done. Line no. 27, Pg no. 1
Line 30: “new-borns” - “newborns”
Response: Done. Line no. 29, Pg no. 1
Line 81: “UL-131” - change for “UL131”
Response: Done. Line no. 89, Pg no. 3
Line 217 “in in early” - remove one “in”
Response: Done. Line no. 167, Pg no. 5
Line 264: “Recently, A few” - lower letter “a”
Response: Corrected. Line no. 203, Pg no. 5
Line 340: remove the large interruption to line 341
Response: Done
Line 374: “used proposed” - remove either of them
Response: Modified the sentence. Line no. 322-324, Pg no. 12
Line 540: “DP;; First” - remove the needless semicolon
Response: Done. Line no. 433, Pg no. 12
Reviewer 3 Report
Thank you for asking me to review the revised manuscript entitled "An update on current antiviral strategies to combat Human cytomegalovirus infection". Considerable improvements have been made to the manuscript although there remains few fundamental flaws.
1) The discussion on valganciclovir in pregnancy is not accurate. Although, it has been postulated that valganciclovir in pregnancy is potentially safe, due to the risk of teratogenicity, it is not used clinically. The medication currently used in this regard is high dose valaciclovir (ref Shahar-Nissan 2020 et al. Lancet)
2) the reference for congenital CMV outcomes needs to include at least the sentinal study by Kimberlin et al 2015 NEJM
3) Unless it simply hasn't been included in the draft provided to me, I cannot see the added figure describing the mechanisms of action and where mutations to each antiviral agent (and associated cross resistance patterns) affects drug activity. I believe this is essential for the reader to understand where in the viral replication cascade the drugs work and thus where resistance in each UL97,54 etc. would limit their utility
4) Although it is difficult to follow with only the tracked changes version, grammatical and sentence,consistency in use of abbreviations, and structuring issues remain.
Author Response
Reviewer 3
Thank you for asking me to review the revised manuscript entitled "An update on current antiviral strategies to combat Human cytomegalovirus infection". Considerable improvements have been made to the manuscript although there remains a few fundamental flaws.
1) The discussion on valganciclovir in pregnancy is not accurate. Although, it has been postulated that valganciclovir in pregnancy is potentially safe, due to the risk of teratogenicity, it is not used clinically. The medication currently used in this regard is high dose valaciclovir (ref Shahar-Nissan 2020 et al. Lancet)
Response: We have now removed this point.
2) the reference for congenital CMV outcomes needs to include at least the sentinal study by Kimberlin et al 2015 NEJM.
Response: We have added the reference. Line no. 141, Pag no. 4, Ref no. 37
3) Unless it simply hasn't been included in the draft provided to me, I cannot see the added figure describing the mechanisms of action and where mutations to each antiviral agent (and associated cross resistance patterns) affects drug activity. I believe this is essential for the reader to understand where in the viral replication cascade the drugs work and thus where resistance in each UL97,54 etc. would limit their utility
Response: In the revised version, we have added the figure. Fig 2.
4) Although it is difficult to follow with only the tracked changes version, grammatical and sentence, consistency in use of abbreviations, and structuring issues remain.
Response: In the revised version, we have tried to remove all these issues.
Author Response
Reviewer 4
General comments:
The reviewer appreciates that in the revised manuscript the authors tried to address the concerns raised during the first reviewing process, e.g. Figure 2 improved greatly. However, there are still a number of points, which should beaddressed before the review is suitable for publication in Viruses. The reviewer is still of the opinion that many sections could be presented in a more structured manner. Also, several references are still missing/incorrect (see specific comments).
Specific comments:
Line 18: should read “chronic state of infection” ïƒ delete “a”
Response: Done. Line no. 18
Line 28: should read “Human cytomegalovirus” ïƒ delete the”
Response: Done. Line no. 27, pg no. 1
Line 28: should read “virus either existing”
Response: Done. Line no. 27, pg no. 1
Line 29: should read “in immunocompromised patients ïƒ add “in”
Response: Added. Line no. 28, pg no. 1
Line 71: should read “particle with a diameter” ïƒ add “with a”
Response: Added. Line no. 75, pg no. 2
Lines 76-77: what is meant by “and to its replication” – viral replication? – please clarify
Response: As suggested we have explained sentence. Line no. 78-79, pg no. 2
Line 77: reference [13] doesn’t seem appropriate as it summarizes HCMV tegument proteins but not glycoproteins
Response: Thank you for your comment, accordingly corrected reference. Line no. 79
Line 81: should read “UL131” ïƒ without hyphen
Response: Correction done. Line no. 82, pg no. 2
Figure 1: Why are there two identical viral particles ïƒ delete the upper one and keep the larger version at the bottom. The glycoprotein complexes should be drawn outside of the lipid envelope (anchored to the outer lipid membrane).Abbreviations like gC-II, TC, gC-I and PC should be explained in the figure legend.
Response: As suggested we have done changes in figure and abbreviations details mentioned in figure legend.
Line 95: should read “HCMV” ïƒ delete “the”
Response: Correction done. Line no. 87, pg no. 3
Line 96: Exchange the term “cells” by virions or virus particles
Response: Correction done. Line no. 88, pg no. 3
Lines 160-161: what is meant by “Viral DNA replication is mediated by ganciclovir-5’-triphosphate (ganciclovir-TP).” ïƒ please clarify
Response: As suggested we have clarified statement in the revised manuscript. Line no. 125-129, pg no. 4
Lines 166-169: Citations are missing
Response: Citation added. Line no. 131, Pg no. 4, Ref no. 34
Lines 177-179: Citation is missing
Response: Citation added. Line no. 132, Pg no. 4, Ref no. 33
Line 217: delete one “in”
Response: Corrections done. Line no. 166, pg no. 5
Lines 217-218: what is meant by “in early haematopoietic stem cell transplant” ïƒ
Response: Statement modified in the revised manuscript. Line no. 167, pg no. 5
Line 238: should read “plays a critical role”
Response: Done. Line no. 183, pg no. 5
Line 239: should read “packaging of” ïƒ add “of”
Response: We have removed the statement.
Lines 239-240: what is meant by “which are transported to the nucleus of the host cell for replication”
ïƒ please clarify
Response: We have modified the statement in the revised manuscript. Line no. 183-185, pg no. 5
Line 242: should read “prevents”
Response: Done. Line no. 187, pg no. 5
Lines 244-245: should read “binding of” ïƒ exchange “with” by
“of”
Response: We have removed the statement.
Line 245: exchange “Pul56” by “pUL56”
Response: Done.
Line 245: should read “reduces”
Response: We have removed the statement.
Line 250: should read: “humans”
Response: Done. Line no 192, pg no. 6
Line 251: should read: “without significant side”
Response: Done. Line no 193, pg no. 6
Line 254: what is meant by “capsids are transported to the nucleus of the host cell for replication” ïƒ please clarify
Response: We have removed the sentence.
Lines 266-267: what do the abbreviations VPD and VCD stand for
Response: We have explained in the revised version. Line no 206, pg no. 5
Line 271: should read “unreported” ïƒ without hyphen
Response: Done. Line no 210, pg no. 5
Line 272: should read “HCMV” ïƒ abbreviations should be consistent throughout the manuscript
Response: Done. Line no 211, pg no. 6
Lines 273-275: The last sentence of this passage does not provide novel information and is identical to the precedingsentence ïƒ please delete
Response: As suggested we have deleted statement in the revised manuscript.
Line 280: reference [47] is incorrect!! ïƒ This publication is a case report where leflunomide failed to control recurrentHCMV infection in the setting of renal failure after allogeneic stem cell transplantation and not after renal transplantation. Effective leflunomide therapy for HCMV disease has been reported in the following publications:https://pubmed.ncbi.nlm.nih.gov/15489872/ and https://pubmed.ncbi.nlm.nih.gov/15167608/.
Response: Thank you for your comment. As suggested we have added correct reference. Line no 219, pg no. 7, Ref no. 54
Line 284: should read „stem cell transplant recipients” ïƒ without “s”
Response: Done. Line no 219, pg no. 7
Lines 289-293: First sentence of this passage needs to be revised.
Response: As suggested we have modified passage in the revised manuscript. Line no. 225- 227, pg no. 7
Table 1: In the major toxicity column of ganciclovir it should read “bone marrow suppression” and bone marrowsuppression of letermovir is missing a “p”
Response: Done correction. Table 1
Line 349: should read: “polyadenylation” ïƒ delete one “yl”
Response: Done. Line no 302, pg no. 9
Line 351: should read “shuts down”
Response: Done. Line no 303, pg no. 9
Lines 351-352: should read “proteins”
Response: Done. Line no 304, pg no. 9
Line 358: in vitro and in vivo should be written in italics throughout the text
Response: Done. Line no. 299, 310, 319 pg no. 9
Lines 372-374: First sentence of this passage needs to be revised.
Response: As suggested we have modified passage in the revised manuscript. Line no 322-324, pg no. 9.
Line 384: should read “and a multiplex” ïƒ add “a”
Response: Done. Line no 333, pg no. 9
Line 402 should read “antivirals” ïƒ add “s”
Response: Done. Line no 363, pg no. 11
Lines 427-429: TALEN-based antiviral strategies have shown promise in laboratory studies with MCMV… ïƒ reference is missing
Response: We have added reference in the revised manuscript. Line no 372, pg no. 11, ref no 85
Lines 434-437: What is meant by aptamers have been used for targeting viral envelopes?ïƒ please clarify
Response: As suggested we have revised the statement in the revised manuscript. Line no 379, pg no. 11
Lines 441-442: should read “(-550 to -750 bp) and (-750 to -1140 bp) ïƒ please add “bp”
Response: Done. Line no 346, 357, pg no. 10
Figure 2 is generally hard to read, pleases enlarge the text.
Response: Done. Fig 3, Page no. 10
Line 446: should read “UL102” instead of UL1O2
Response: Done. Line no 351, pg no. 10
Line 451: The reviewer would not call figure 2 a diagram but e.g. a schematic illustration.
Response: Done. Line no 354, pg no. 10
Line 460: should read “reduction of viral plaque formation/count/numbers”
Response: Done. Line no 387, pg no. 11
The reviewer thinks that paragraph “4. Cytomegalovirus vaccines” should be switched to 3. so that “Possible alternativestrategies to combat HCMV infection” represents the last paragraph of the manuscript as this is the major focus of thisreview.
Response: Thank you for your suggestion, we have switched paragraphs accordingly. Line no. 232- 258.
Line 492: should read “neutralizing antibodies”
Response: Done. Line no 245, pg no. 8
Line 500: should read “RNA-based”
Response: Done. Line no 251, pg no. 8
Lines 497-501: Reference is missing for the first sentence of this passage.
Response: Reference added. Line no 248, pg no. 7, ref no. 62
Lines 502-503: “Initially, gB seemed a perfect choice, but trails showed limited efficacy.” ïƒ reference is missing
Response: Reference added. Line no 251, pg no. 7, Ref. 63
Line 504: exchange “the” HCMV vaccine by “an” HCMV vaccine
Response: Done. Line no 254, pg no. 7
Line 505: should read “into host cells”
Response: Done. Line no 255-256, pg no. 8
Line 530: ensure uniform spelling of “in vitro” and “in vivo” throughout the manuscript
Response: Done. Line no. 207, Pg no. 5